# Seismological expression of the iron spin crossover in ferropericlase in the Earth's lower mantle

Grace E. Shephard [1✉], Christine Houser [2], John W. Hernlund[2], Juan J. Valencia-Cardona [3], Reidar G. Trønnes [1,4] & Renata M. Wentzcovitch [5,6,7✉]

The two most abundant minerals in the Earth's lower mantle are bridgmanite and ferropericlase. The bulk modulus of ferropericlase (Fp) softens as iron d-electrons transition from a high-spin to low-spin state, affecting the seismic compressional velocity but not the shear velocity. Here, we identify a seismological expression of the iron spin crossover in fast regions associated with cold Fp-rich subducted oceanic lithosphere: the relative abundance of fast velocities in P- and S-wave tomography models diverges in the ~1,400-2,000 km depth range. This is consistent with a reduced temperature sensitivity of P-waves throughout the iron spin crossover. A similar signal is also found in seismically slow regions below ~1,800 km, consistent with broadening and deepening of the crossover at higher temperatures. The corresponding inflection in P-wave velocity is not yet observed in 1-D seismic profiles, suggesting that the lower mantle is composed of non-uniformly distributed thermochemical heterogeneities which dampen the global signature of the Fp spin crossover.

[1] Centre for Earth Evolution and Dynamics (CEED), Department of Geosciences, University of Oslo, Oslo, Norway. [2] Earth-Life Science Institute, Tokyo Institute of Technology, Tokyo, Japan. [3] Logic Technology Development, Intel Corporation, Hillsboro, OR, USA. [4] Natural History Museum, University of Oslo, Oslo, Norway. [5] Department of Earth and Environmental Sciences, Columbia University, New York City, NY, USA. [6] Lamont-Doherty Earth Observatory, Columbia University, Palisades, NY, USA. [7] Department of Applied Physics and Applied Mathematics, Columbia University, New York City, NY, USA. ✉email: grace.shephard@geo.uio.no; rmw2150@columbia.edu

Mineral physics experiments[1–3] and theory[4,5] consistently predict that the d-electrons of $Fe^{2+}$ in Fp, (Mg,Fe)O, change from a high-spin to low-spin (HS-LS) state at mid-lower mantle conditions (Fig. 1a). However, the compressional velocity of a homogeneous pyrolitic mantle (Fig. 1d) does not fit global seismic reference profiles when the effects of the iron spin crossover are included in the predicted seismic velocity computations[6]. Confirmation of the existence and observation of the iron spin crossover in the Earth's mantle is relevant because it is expected to alter material properties such as density, viscosity, elasticity, thermal conductivity and elemental partitioning in the lower mantle[7]. The rheological consequences of such material changes mean that subducted slab, mantle plume and deep mantle dynamics are affected by the crossover, including reduced viscosity, enhanced vertical flow and slab stalling[8–10]. In spite of its potential importance, the spin crossover in Fp has thus far eluded seismological detection, suggesting that the predictions are inaccurate, the signature is below the detection threshold, and/or the lower mantle has a lower (Fe+Mg)/Si ratio than the shallower mantle.

The distinct effects of the Fp spin crossover on compressional (P-wave) and shear (S-wave) velocities offer a promising target for geophysical observation (Fig. 2). In particular, a volume reduction during the spin crossover[4] inevitably increases the compressibility of Fp and decreases its bulk modulus[11]. Both the onset pressure and the pressure interval of the mixed-spin region (where both high- and low-spin states coexist) are predicted to increase at higher temperatures[7,11–13]. The temperature dependence of the pressure onset and pressure range of the HS-LS crossover results in an anomalous dependence of the bulk modulus on temperature[14], with little influence on the shear modulus (Fig. 2a). The significance of this effect increases for higher iron contents and abundance of Fp[7]. Specifically, the transition pressure does not change significantly for FeO concentrations below 20 mol% in Fp, which is representative for the lithological range from fertile peridotite to harzburgite[15]. The magnitude of our predicted velocity reductions are supported by experimental data[5], especially from Brillouin scattering[13,16], and motivate investigation of thermal anomalies in the lower mantle.

One consequence of the bulk modulus softening during the iron spin crossover in Fp is that the P-wave velocity is generally reduced while the S-wave velocity remains unaffected. While pyrolitic model compositions have been found to be consistent with 1-D seismic models[17,18] (i.e. global seismic reference/average profiles), Fig. 2 demonstrates that the Fp content of pyrolitic rocks is sufficient to significantly reduce P-velocity across the mid-mantle iron spin crossover when the effects of the iron spin crossover are included. The velocities are computed using ab initio mineral physics calculations[15] on a simplified pyrolite composition listed in Table 1. The P-wave and S-wave velocities for the full range of Fp-bearing rocks (from Fp-free model composition to Fp-rich harzburgite, Supplementary Fig. 1) reveal the influence of variable Fp abundances on mid-mantle seismic velocities. Most domains in the lower mantle likely contain Fp in the range bounded by these two compositions. Figure 2 and Supplementary Fig 1 reveal that mid-mantle thermal anomalies in Fp-bearing rocks are predicted to produce a measurable S-wave seismic anomaly while the predicted P-wave seismic anomaly would be greatly diminished. Thus, we survey P-wave and S-wave velocity tomography models for the distinctive seismic signal of the iron spin crossover in Fp: S-velocity anomalies produced by lateral temperature variations persist but P-velocity anomalies weaken within the expected mixed-spin region (see Fig. 2c).

In the lower mantle, regions with fast seismic anomalies are commonly interpreted as cold, sinking oceanic lithosphere[19,20]. These subducted slabs typically comprise 5–7 km of basaltic crust underlain by ~60–80 km of mantle. The lithospheric mantle is depleted in Si due to the extraction of basaltic melt, increasing its Mg/Si ratio and thus the relative amount of Fp in the lithospheric mantle (at lower mantle conditions). Because the magnitude of the velocity reduction due to the iron spin crossover depends on the abundance of Fp (Supplementary Fig. 1), these fast seismic regions should host the strongest spin crossover-related seismic signals. Slow regions in the mid-mantle may be due to return flow of former oceanic plates after becoming warm and buoyant near the core-mantle boundary[21]. Depending on wavelength and depth, previous comparisons of P- and S-wave tomography models suggest different seismic characteristics in the lower mantle, including changes in the ratio and correlation of S- and P-wave velocity variations and apparent disruption of imaged fast/slow features[22–26]. Direct P- and S-waves and additional phases such as PP, PP-P, SS and SS-S provide nearly uniform coverage across the mid-mantle[27]. Hence, both P-wave and S-wave models are well-resolved vertically and laterally in the mid-mantle[28,29]. Given the similarities in P- and S-wave

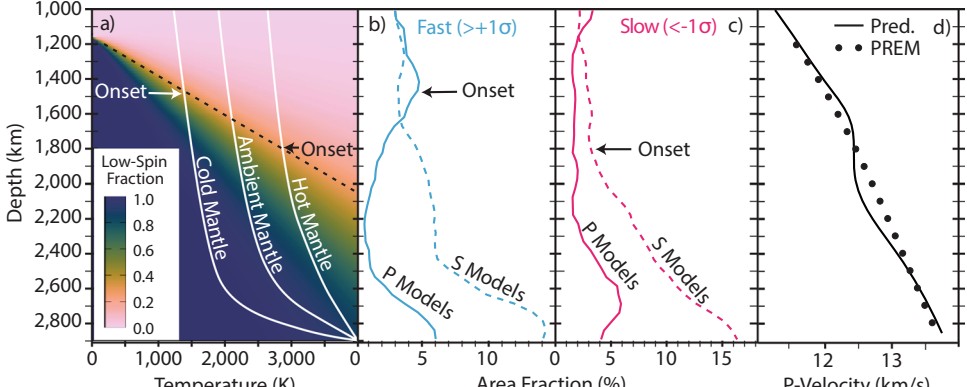

**Fig. 1 Comparison of mineral physics predictions and seismological observations of the Fe spin crossover in Fp. a** The depth-temperature distribution of low-spin Fe and three schematic mantle geotherms. The onset of the mixed-spin region occurs at shallower depths for relatively colder temperatures and at greater depths and over a broader range at higher temperatures. Surface area of fast (**b**) and slow (**c**) velocities in the lower mantle imaged by unanimous consensus (4/4 models) corresponding to the vote maps shown in Figs. 5 and 6. A divergence in the seismic signal between P-wave models relative to S-wave models is revealed below the respective "onset" depths of ~1400 and 1800 km. **d** The calculated profile of P-wave velocities for pyrolite[15] (see Supplementary Fig. 9) reveals a significant departure suggesting that the signature of the spin crossover is not a globally averaged feature. See Methods. Colour gradients from ref. [73].

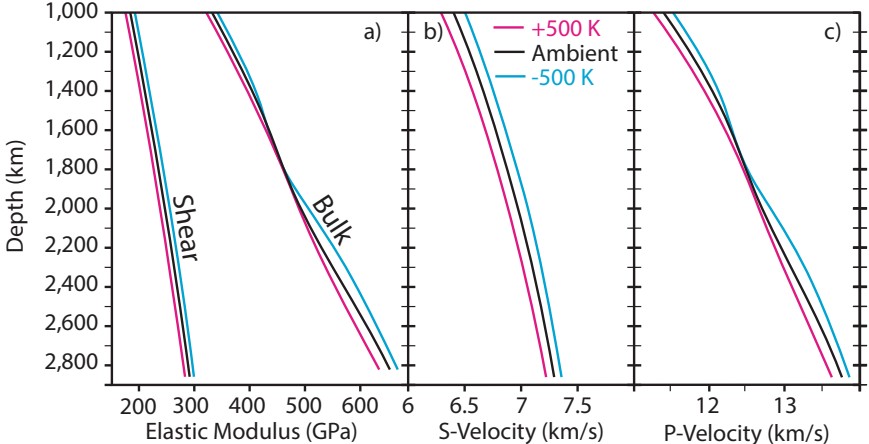

**Fig. 2 Depth dependence of elastic moduli and seismic velocities for pyrolite composition listed in Table 1[15]. a** Shear and bulk moduli. The anomalous behaviour of the bulk modulus, K[11] ($K = -V(dP/dV)$, where $V$ is volume and $P$ is pressure) is due to the iron octahedron volume collapse during the spin-state change[4] associated with the spin crossover. As previously shown[11], the pressure onset, the pressure range of the HS-LS crossover, and associated anomalies in K are temperature dependent (same legend all panels). The shear modulus, G, ($G = \tau/\gamma$, where $\tau$ is the shear stress and $\gamma$ is the shear strain) is not significantly affected by the octahedron volume reduction and increases monotonically with increasing pressure and decreasing temperature. **b** S-wave ($V_s = \sqrt{G/\rho}$) and **c** P-wave ($V_p = \sqrt{(K + 4/3G)/\rho}$) velocity for pyrolite at different temperatures. The average mantle temperature profile (black) is calculated by setting the starting temperature to 1873 K at 660 km, and integrating the adiabatic temperature gradient[15] through the lower mantle. The blue/magenta curves were calculated using the same technique, but decreasing/increasing the temperature at 660 km by ±500 K. In the mixed-spin region, the softening of K[11] causes a reduction in P-wave velocity sensitivity to isobaric temperature variations[14], while the S-wave velocity remains sensitive to such temperature variations. Consequently, the expected seismic signal of iron-bearing Fp in the lower mantle consists of a disruption of vertically coherent thermal structures in P-velocities, whereas a coherent thermal structure is apparent in the S-velocities[12,14]. Additional compositions shown in Supplementary Fig. 1 demonstrate the calculated seismic velocities dependence on Fp proportion.

**Table 1 Compositions in mol.% used in this paper expressed as Bm ($Mg_{1-x-z}Fe_xAl_z$)($Si_{1-z}Al_z$)$O_3$ + Fp, ($Mg_{1-y}Fe_y$)O + Ca-perovskite (Cpv), $CaSiO_3$.**

|              | Bm    | Fp    | Cpv  | x      | y      | z      |
|--------------|-------|-------|------|--------|--------|--------|
| Harzburgite  | 55.99 | 42.74 | 1.27 | 0.0693 | 0.1315 | 0.0149 |
| Peridotite   | 60.4  | 34.42 | 5.19 | 0.0747 | 0.1458 | 0.0492 |
| Pyrolite     | 62.93 | 31.58 | 5.49 | 0.0779 | 0.1523 | 0.0553 |
| Fp-free model| 94.18 | 0.0   | 5.82 | 0.1220 | 0.0    | 0.0453 |

These simplified compositions were also used in ref. [15]. The simplified peridotite composition based on refs. [75,76] is included to demonstrate its similarity to pyrolite used in Figs. 1–2. In our calculations we used a partitioning coefficient $K_D = [Fe/Mg]Bm / [Fe/Mg]Fp = [x/(1-x-z)] / [y/(1-y)] = 0.5$.

resolution in the mid-mantle, we focus on characteristics within fast and slow velocity regions in P-wave and S-wave models using both individual tomography models and combined through tomographic vote maps.

Here, we identify the seismological expression of the iron spin crossover in the mid-lower mantle by investigating fast and slow regions in P- and S-wave tomography models. The identified signal (below ~1400 km in fast and below ~1800 km in slow regions) corresponds to mineral physics predictions, including a temperature dependence of the pressure onset and pressure range of the high-to-low spin crossover in Fp.

### Results and discussion

**Individual tomography models.** With the rapid expansion of data acquisition and improvements in inversion and modelling techniques in recent decades, numerous global P- and S-wave seismic tomography models are now available[30–32]. Here we use four P-wave models (DETOX-P01[33], GAP-P4[34,35], HMSL-P06[28], MITP-2011[36]), and four S-wave models (HMSL-S06[28], S40RTS[37], savani[38], SEMUCB-WM1[39]) that capture a range of global, whole mantle tomographic data and techniques. Supplementary Fig. 11

shows the results for two additional tomography models. Figure 3 shows the overall methodology we use to determine the vertical and lateral extent of fast and slow regions. The percentage of the surface area at each depth that is identified as a fast (or slow) seismic anomaly is calculated as derived from a contour threshold. We define the threshold for fast/slow as seismic wave speed anomalies that deviate by more than one standard deviation ($\sigma$) in the central portion of the seismic velocity distribution over a reference depth range of 1000–2200 km (i.e. $\geq 1\sigma$ for fast, $\leq -1\sigma$ for slow, see Methods). Supplementary Figures 2-4 show the resiliency of alternative thresholds for defining fast/slow anomalies of the depth-dependent trends.

The surface area of fast and slow wave speed anomalies, plotted at depth intervals of 50 km across the lower mantle, reveals distinct patterns between P-wave and S-wave models (Fig. 4), including that there is more variability between the P-wave models than between the S-wave models, as also noted in ref. [40]. Furthermore, when viewed collectively (Fig. 4), these individual tomography models indicate that the area covered by fast P-wave velocities decreases in the lower mantle across the depth range of ~1400–2200 km relative to S-wave velocities. The overall trend of fast areas in the S-wave models increases with depth (from ~15% coverage at 1200 km depth to ~40% coverage near the core-mantle boundary).

**Vote maps.** The differences in the characteristics and distribution of seismic anomalies, both within and between the P-wave and S-wave models, and as illustrated in Fig. 4, lead us to further inspect the geographic similarity of these patterns. Thus, we employ a vote map method to examine the surface area of fast and slow regions that are common to all of these models[41] (Fig. 3a–c, see Methods). The highest count (4 votes; Fig. 1b–c) indicates regions where all four models (for P-waves or S-waves) agree on the existence of fast anomalies (i.e. $\geq 1\sigma$) at a given depth interval. For example, the pattern of fast anomaly vote maps

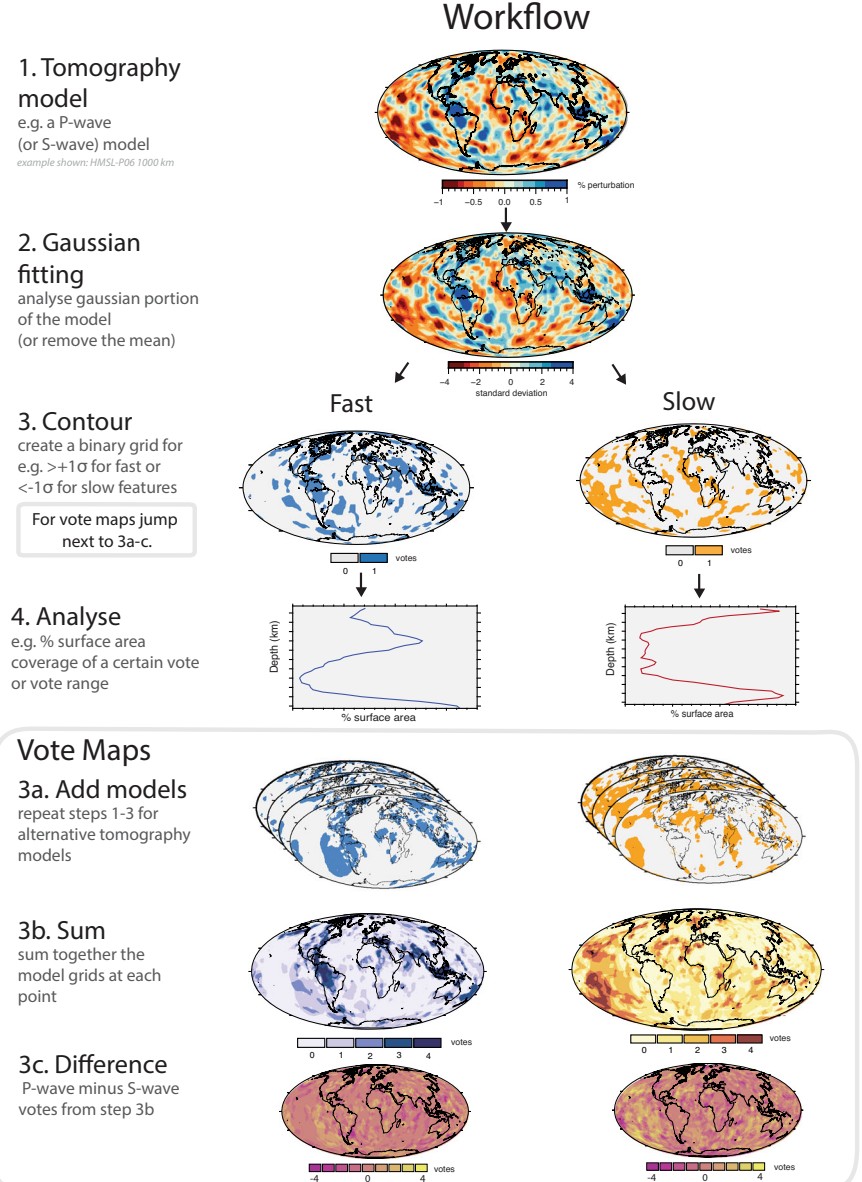

**Fig. 3 Workflow used in this paper.** Step 1: select tomography model. Step 2: a gaussian fitting procedure is applied (see Methods). Step 3: a contour metric is chosen e.g. fast (>+1σ from the mean), and slow (<−1σ from the mean) and binary grids are created. Step 4: surface area coverage of the contour is extracted across the lower mantle. For the vote map methodology additional steps are applied between Steps 3 and 4. Step 3a: binary grids from additional tomography models are created as per Steps 1–3. Step 3b: at each grid point the constituent models (here 4 models) are summed. Step 3c: difference vote maps can be generated, for example by subtracting the S-wave votes from the P-wave votes.

(Fig. 5) reveals north-south trending subducted slabs under North America, related to long-lived subduction along eastern Panthalassa, as well east-west trending slabs related to palaeo-subduction within the Tethys Ocean[41]. Likewise, the cluster analysis-based vote map technique[40,42] illuminates the common morphology of large low shear velocity provinces (LLSVP) in the lowermost mantle.

The vote map procedure does not add any features not already present in the constituent tomography models; it rather highlights the features that are common to multiple models. For example, when analysing global, whole mantle models, it is difficult to know if any given patch of anomalously fast or slow velocity is due to imperfect input data, inversion artefacts, or is a genuine signal present in the lower mantle. Each tomography model utilizes different types of input data, parameterization, regularization and other subjective choices in their construction (e.g. see model

compilations[25,32,41,43]). Therefore, we expect that features introduced into individual tomography models due to unique tomographic data and/or modeling approaches will not have a strong influence in the highest count of the vote maps (e.g. ref. [42]). Furthermore, our contour analysis is global and is therefore less susceptible to locally restricted seismic anomalies. By the same token, vote maps offer a framework for unique, localized anomalies to be identified and evaluated between tomography models.

Similar to the individual models shown in Fig. 4, the surface area of fast velocity regions in P- and S-wave vote maps (Fig. 1b) diverges in the mid-mantle beginning at ~1400 km depth. This observed depth is similar to that predicted for the mixed-spin region in Fp for a pyrolitic composition, Fig. 1a. The decorrelation of anomalous P-wave and S-wave abundances is a robust signal regardless of the analysis type, contour threshold, highest

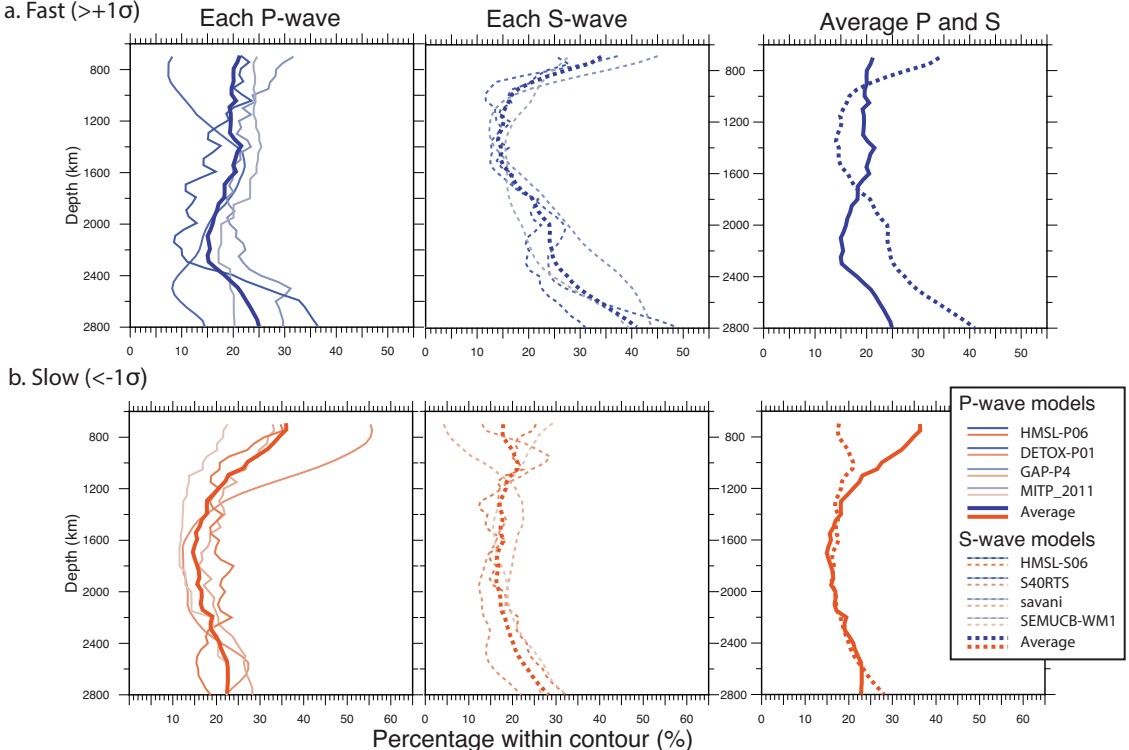

**Fig. 4 Changes in surface area for the individual tomography models.** Depth-dependent changes in surface area for the **a** fast (>+1σ from the mean), and **b** slow (<−1σ from the mean) regions, as applied to each of the 4 P-wave, and 4 S-wave tomography models. The individual model names and details are shown in Supplementary Figs. 2-3. The bold lines are the average of each panel, which are extracted and shown in the rightmost panels. This procedure is similar to that shown in Fig. 1b, c except without the vote map analysis.

vote counts or model combinations. For example, we also test the influence of sequentially adding in the tomography models (from 1 to 4 models) in Supplementary Fig. 5, as well as alternative combinations of 3 of the 4 models in Supplementary Fig. 6. The decorrelation signal of fast P-waves and S-waves was previously examined[41] using an expanded set of 14 tomography models. Furthermore, in Supplementary Fig. 7 we compare the depth-dependent signal for lower vote counts rather than just the maximum vote count of 4 in Fig. 1b, c. These tests demonstrate the robustness of the P- versus S-wave signal.

To further test the hypothesis that the decoupling of the fast velocity signals in P-wave and S-wave models reflects a spin crossover in Fp, we examine seismically slow, and presumably warm, portions of tomographic models. The surface area of slow velocity regions in the vote maps (Fig. 1c) reveals that the onset of divergence shifts to greater depths (i.e. below ~1800 km) consistent with the deepening of the spin crossover as temperature increases (Fig. 1a). It also demonstrates the predicted broadening of the spin transition at higher temperatures (Fig. 1a), leading to a more diffuse seismic signal. The divergence in slow velocity regions between P- and S-wave models is likewise more subtle than that observed in fast velocity regions, see also Supplementary Figs. 5-6. The vote maps are a useful tool for examining these types of subtle but consistent and geographically coherent velocity anomaly patterns, i.e. the common signal derived from slow anomalies in the individual tomography models (Fig. 4) is more discernible in the vote maps (Fig. 1b, c).

Figures 5–6 show horizontal and vertical cross-sections, respectively, through vote maps of fast and slow velocity regions. All vote values are shown in these images (i.e. 0–4 votes). We can qualitatively observe a divergence in the agreement among P-wave models compared to S-wave models in the mid-mantle

(see 1500 km depth) for both fast and slow anomalies. For example, fast anomalies attributed to subducted slabs under North America and SE-Asia appear contiguous through the mid-mantle in S-wave vote maps. However, the P-wave vote maps show a weakening of the fast seismic signal in the mid-mantle (coherent fast anomalies at the top and bottom of the lower mantle are still observed). Likewise, the slow mantle domain under Africa and the Afar hotpot region is more readily apparent in S-wave than in P-wave vote maps towards the lowermost mantle (below ~2000 km depth). This observed interference is what is predicted for the spin crossover in Fp[14]. The disruption of slab and plume images in the mid-mantle has also been noted in other studies[23,26,44–47]. Figure 7 shows difference maps, highlighting the spatial and depth-dependent differences between the P- and S-wave vote maps. There is a dominance of S-wave votes over P-wave votes in the lowermost mantle, especially for the fast anomalies in regions of long-lived subduction (Figs. 5 and 6), and in the slow anomalies for regions corresponding to the LLSVP domains. Note that our definition of anomalous material is based on characteristic behaviours in the depth range 1000–2200 km (see Methods), and any discussion of anomalies outside of this depth range should be considered relative to these standards.

We note that the abundance of fast velocity regions in both P-wave and S-wave models (Figs 1b, 4) increases between ~2500–2800 km depth. This could reflect a build-up of slab material and/or a change in the globally averaged subduction flux, as estimated from mantle sinking rates[41,48]. Furthermore, the area of fast S-wave anomalies increases more rapidly toward the base of the mantle than the P-wave anomalies. This relative difference could (in addition to potential slab-volume changes) be caused by the appearance of post-perovskite in cold mantle, which is expected to increase S-wave velocity by ~1–2% but has notably

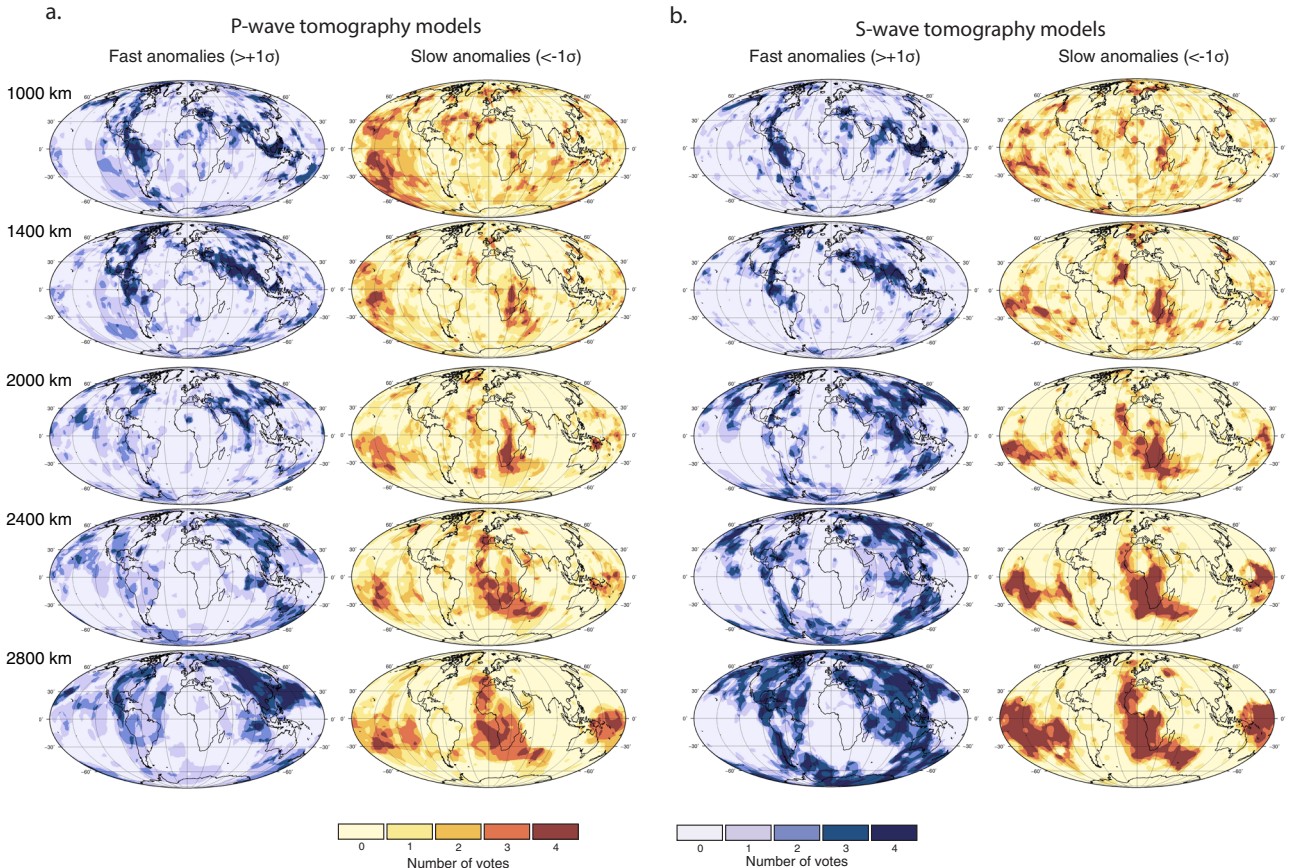

**Fig. 5 Vote maps of mantle tomography models.** Variable depth vote maps for fast (>+1$\sigma$ from the mean), and slow (<−1$\sigma$ from the mean) mantle for all **a** 4 P-wave and **b** 4 S-wave tomography models (see Methods) considered here. In general, lower mantle vote maps of fast velocity regions reveal subducted slabs, and slow maps reveal plumes and antipodal large low shear wave velocity provinces (LLSVPs). In this map view, the P-wave model vote maps appear less coherent than the S-wave model vote maps in the fast velocity regions of the mid-mantle. Colour gradients from ref. [73].

smaller effect on P-waves[49–51]. The difference maps between P-wave and S-wave votes at 2800 km depth, shown in Fig. 7, reveal regions of potential post-perovskite (dark purple regions indicating higher S-wave votes than P-wave votes). The antipodal LLSVPs, beneath the Pacific and Africa, are areas where the S-wave velocity is more strongly reduced relative to the P-velocity. The divergence between S- and P-wave behaviours within the LLSVPs has been used to argue for the presence of dense chemically distinct piles in these regions[52]. Below ~2800 km we expect the trends to be complicated by core-mantle boundary layer dynamics[40,53,54].

**Effect of iron partitioning.** The spin crossover is expected to influence the iron partitioning ($K_D$) between bridgmanite (Bm), $(Mg,Fe)(Al,Si)O_3$, and Fp[1,55] by increasing the iron concentration in Fp, thus decreasing $K_D$[55]. Since the spin crossover may shift to higher pressures as $K_D$ decreases[7], the pressure range over which the crossover occurs could be wider than that predicted for the constant partitioning value of $K_D = 0.5$ which we assign in our calculations. We also investigated how the onset and depth range of the spin crossover changes with variable $K_D$[55] values (Supplementary Fig. 8). The velocities with variable $K_D$ are barely distinguishable from the constant $K_D$ case due to a very weak dependence on the crossover depth on FeO concentrations below ~20 mol.% (e.g. refs. [56,57]). This prediction is consistent with our observation that the effects of the crossover on P-velocities in fast velocity regions does not extend below ~2500 km depth even for Fp-rich compositions, Supplementary Fig. 1.

**Implications for lower mantle composition.** A reduction in the area of mid-mantle fast velocity regions in P-wave models compared to S-wave models is consistent with the predictions of an iron spin crossover in Fp. To investigate the 1-D signal, we compare our predicted P- and S-wave velocities for pyrolite to the Preliminary Reference Earth Model (PREM)[58] radial P-wave profile, Fig. 1d. Due to uncertainties in the average mantle geotherm[6], we calculate the temperature profile which aligns our predicted S-wave velocity to PREM (Supplementary Fig. 9). We demonstrate that the different behaviour of P- and S-wave velocities during the spin crossover could be detected by 1-D seismic profiles for a uniform Fp-bearing mantle, where S-wave velocity is a proxy for temperature, since it is not affected by the iron spin crossover. However, the intensity of the P-wave velocity reduction depends on the Fp abundance and the abundance of iron in the Fp, which could be different than our model composition. Since Bm has a similar spin crossover for ferric iron ($Fe^{3+}$) located in the octahedral B-site[59], the lack of a distinct iron crossover signal in 1-D seismic profiles[60] suggests that aluminium replaces a majority of the $Fe^{3+}$ on the B-site[61]. While there are still uncertainties with the spin crossover pressure range and magnitude of the seismic signal at relevant mantle temperatures, the general agreement between theoretical predictions[5] and experimental measurements[13] at room temperature compels us to consider how to reconcile the patterns observed here, in both the fast and slow regions, with those from the 1-D average profiles.

The observed divergence in P-wave and S-wave models in the fastest and slowest regions of the mid-mantle indicates that the iron spin crossover signal is seismically detectable in the deeply subducted Fp-enriched oceanic lithosphere. Variable composition,

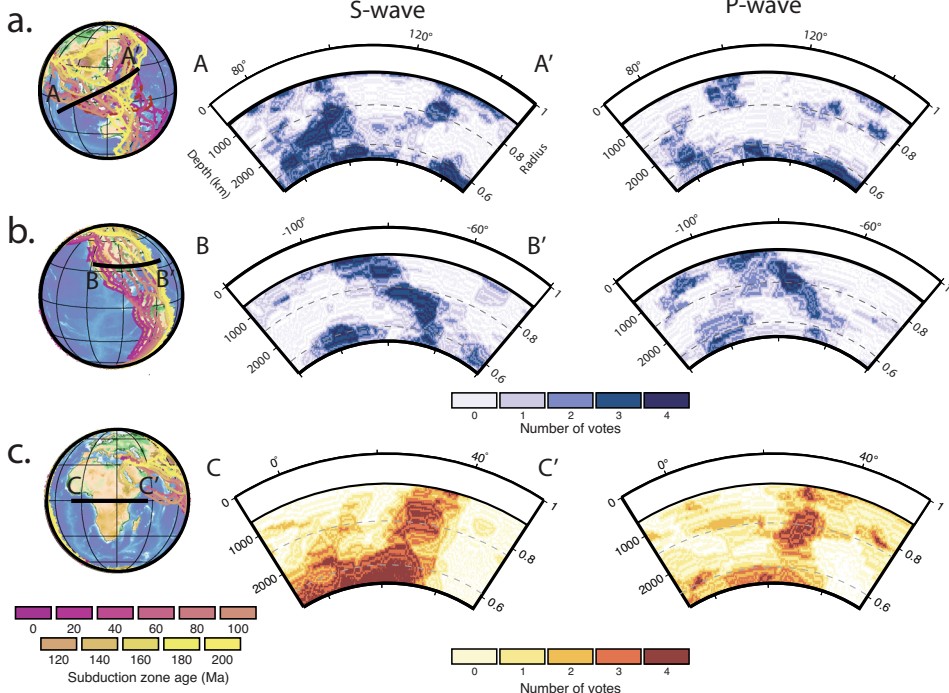

**Fig. 6 Vertical cross-sections through the vote maps.** All four S-wave models (left) and four P-wave models (right) are used for profiles through (**a**) SE Asia and **b** the Farallon slab regions (vote maps for fast velocities), and (**c**) the African-Afar large low shear wave velocity province (LLSVP) and plume region (vote maps for slow velocities). Location maps are overlain with palaeo-subduction zones between 0 and 200 Ma as extracted from a global plate reconstruction[77]. Panels (**a**) and (**b**) identify vertically coherent fast regions that are common to S-wave models in the mid-mantle indicating the presence of subducted slabs whereas the slab signal is weak amongst P-wave models. This is more clear in the fast rather than (**c**) slow regions because at higher temperatures the HS-LS crossover shifts to greater depths where the seismic signal is dominated by the LLSVPs, and also occurs over a broader depth range.

such as regions with less Fp or less iron in the Fp, could disrupt and reduce the spin crossover signal[6,12]. Variations of lateral temperature with pressure introduce a depth dependence to the P-wave seismic velocity inflection (i.e. causing a potential vertical smearing of the signature), which could reduce the globally averaged signal. Geodynamically, the lack of a ubiquitous signal of the iron spin crossover in the lower mantle may indicate that sluggish mantle mixing has yet to erase large-scale heterogeneities. Suppression of a global (1-D) signature of the spin crossover might be explained if significant portions of the lower mantle are depleted in Fp (i.e. enriched in bridgmanite), while Fp-bearing rock that circulates between the shallow and deep mantle is concentrated in upwelling and downwelling channels. This situation is similar to the bridgmanite-enriched ancient mantle structures (BEAMS) model[21] in which Fp-poor regions exhibit a higher viscosity and resist convective mixing[62]. In addition, 1-D seismic models using low-order polynomial parameterizations of radial velocities may have smoothed the seismic signal of the iron spin crossover, which becomes more subdued as the shape and distribution of thermochemical provinces becomes more varied. High resolution regional 1-D profiles are challenging to construct but could reveal the presence or absence of Fp outside of regions with large lateral temperature variations such as those identified in this study. It is possible that more complicated combinations of thermal, phase, and/or chemical composition variations could provide an alternate explanation for Vs-Vp decorrelation. However, the pressure-temperature dependence of the Fp spin crossover provides a unified explanation for why this occurs at these particular distinct depths in slow and fast regions, and does so without the need to invoke depth-dependent changes in the chemical composition of downwelling and upwelling materials.

Our study qualitatively identifies the predicted seismic expression of the iron spin crossover in lower mantle Fp in fast and slow regions of global tomography models. The effects of the iron spin crossover on seismic velocities are subtle[63] but most discernible in the presence of lateral temperature variations in Fp-bearing regions that extend across the depths that span the iron spin crossover range. Our detection of Fp in the fastest and slowest regions without a strong signal in the global average mantle suggests the presence of a mosaic of large-scale thermal and compositional domains in the lower mantle.

## Methods
**Predictions of the spin crossover in Fp.** Figure 1a was calculated for $Mg_{1-y}Fe_yO$ with $y = 18.75\%$, also labelled $Y_{Fe}$ in Eq. (1), using published models[5,14]. The low-spin fraction is calculated as:

$$n(P,T) = \frac{1}{1 + m(2S+1)\exp\left[\frac{\Delta G_{HS-LS}^{st+vib}}{Y_{Fe}k_B T}\right]} \tag{1}$$

where $m = 3$ for the three possible orientations of minority electron d orbital (xy, yz or zx) and $S = 2$. $\Delta G_{HS-LS}^{st+vib}$ includes only static and vibrational energy, without electronic entropy. Our elastic moduli and densities are also taken from published models[5,15,64,65]. These calculations used the rotationally invariant local density approximation $LDA+U_{SC}$ method calculated with a self-consistent Hubbard $U$[66]. Results for Fp were obtained using a 64-atom supercell with an iron concentration of $y = 0.1875$ (24 Mg, 32 O and 6 Fe maximally separated from each other). Thermo-elastic properties were then obtained for other concentrations by linearly interpolating between $y = 0$ and $y = 0.1875$. The vibrational density of states (VDOS) was computed using the vibrational virtual-crystal model[11], and then used in conjunction with the quasiharmonic approximation to predict high temperature effects. The magnitude of the predicted effects are in good agreement with experimental measurements[13]. Thermoelastic properties from[64,65] are used for $Fe^{2+}$- and Al-bearing bridgmanite. While our calculations do not include $Fe^{3+}$, the velocities for $Fe^{3+}$-bearing Bm[65,67] are similar and thus do not effect our estimates of the temperature dependence of the iron spin crossover in ferropericlase. Here we consider the effects of a spin crossover in Fp, proposed spin changes in Fe in Bm are not considered.

## Difference (P-wave minus S-wave votes)

**Fig. 7 Difference maps between P-wave and S-wave models.** Difference maps for the fast (left panels) and slow (right panels) vote maps constructed by subtracting the S-wave model votes (constructed with 4 models) from P-wave model votes (constructed with 4 models). Yellow areas indicate high votes (i.e. the identification of robust features) in P-wave models but not S-wave models, darker purple areas indicate high votes for S-wave models relative to P-wave models, and central colours indicate regions of mutual agreement.

The mantle is thought to be a lithological mix of depleted peridotite and separate slivers and lenses of recycled oceanic crust (ROC). Whereas the current subduction zones have an average ROC to depleted peridotite ratio of about 1/10, this ratio has probably decreased steadily from larger ROC fractions in the Archean[68]. Although a depleted peridotite has higher Fp-content and lower Fe/Mg ratio than a fertile (pyrolitic) peridotite[55], it may have a slightly stronger spin crossover signal. This effect will be weakened by the diluting ROC content.

The shear and bulk modulus (Fig. 2a), S-wave (Fig. 2b) and P-wave velocity (Figs. 1d, 2c) are calculated for a pyrolite[15] composition. The self-consistent geotherms are calculated by setting the starting temperature of the calculation at the top of the lower mantle to 1373 K (the $-500$ K case), 1873 (the average case) and 2373 K (the $+500$ K case) (Supplementary Figs. 8 and 9). The mineralogy of four different bulk composition models are reported in Table 1. They are the same as those derived in ref. [15]. The Earth's actual mantle geotherm is uncertain, and fitting PREM requires non-adiabatic gradients and/or variations in composition with depth in the lower mantle[69]. Here the P-wave velocity in Fig. 1d is calculated using temperatures that align pyrolite S-wave velocity to PREM[58] at each depth,

thus highlighting the inability to match both S-wave and P-wave constraints simultaneously when the effects of a spin crossover are included (S-wave velocities are not significantly affected by the spin crossover, Fig. 2b). Note, that this calculation is performed for a uniform chemical composition without lateral temperature variations.

**Fast and slow mapping.** Most tomography models agree on the large-wavelength structure of seismic velocity anomalies in the mantle e.g. refs. [37,42,44]. However, the amplitudes of the anomalies, as well as smaller scale-features, such as individual subducted slabs, sometimes vary owing to differences in parameterization, data processing, regularization and other subjective influences[23]. There are also few joint P- and S-wave models that do not include some type of scaling of the P-wave model to the S-wave model in the lower mantle. Finally, wide disparities in amplitudes have been noted between various models[43], which may reflect volume and coverage of data, choice of regularization parameters, volume discretization and other influences (Supplementary Fig. 10).

Instead of relying on particular P- and S-wave model pairs and their stated amplitudes, we set a uniform grid in the mantle and generate a contour based on a given positive/negative $\sigma$ deviation for fast/slow anomalies, respectively. We define $\sigma$ for each model by combining and binning all values between 1000 and 2200 km depth (by area) to produce a mid-mantle reference distribution. We use only the mid-mantle to avoid more complex behaviours that appear in the shallow and deep portions of the lower mantle. We then perform an iterative Gaussian fitting procedure[43] on the mid-mantle reference distribution, utilizing values only within the interval $-1 < \sigma < 1$. This procedure avoids the influence of more extreme velocity values on determining the threshold, as these may exhibit non-normal characteristics that would otherwise exert a large influence on the usual arithmetic measure of standard deviation (Supplementary Fig. 10). If the distribution is perfectly normal, then the $\sigma$ value obtained using this procedure would be identical to the usual measure of standard deviation. Note that this procedure also excludes what will come to be defined as "fast" or "slow" anomalies from the determination of what constitutes fast or slow velocities; only the dominant modal variation of relatively modest velocity variations is used to obtain $\sigma$ for each model. Supplementary Fig. 10 shows the mid-mantle reference distributions for all of the models used in this study, along with the Gaussian fits we obtain in the $-1 < \sigma < 1$ intervals.

The differing extent of mid-mantle fast anomalies in Vp and Vs requires the simultaneous effect of temperature, composition, and phase. Any temperature effect alone would be expected to manifest in Vs and Vp to a similar extent and relative magnitude. Since fast anomalies are often interpreted as cold subducted oceanic lithosphere, the next property to consider is composition. Subducted slabs are composed of a thin basaltic crust and a Fp-rich mantle lithosphere. The thin basaltic crust may host anomalous seismic characteristics, but these seismic anomalies are not expected to be detected by seismic tomography at scales of 1000's of km in the mid-mantle. If Fp exists in higher quantities in fast velocity regions, then the bulk rock would have slightly reduced velocity since Fp has slower Vp and Vs than Bm. However, the fast velocity anomaly observed in seismic tomography indicates that temperature dominates the seismic signal even in these more Fp-rich subducting slabs. Since variations in Fp concentration have a similar effect on Vp and Vs, it is difficult to decouple Vp and Vs even with temperature and composition. Fp concentration in slabs opens up the possibility of the iron spin transition which reduces Vp and not Vs in the presence of a thermal anomaly.

**Vote maps**. Vote maps are a simple tool, developed to detect the existence and map the distribution of material exhibiting particular seismic characteristics in the lower mantle. Lekic et al.[42] and Cottaar and Lekic[40] developed a k-means cluster analysis-style vote map for the lower mantle. This was expanded in Shephard et al.[41], who developed an alternative vote map technique using tomography and depth-dependent metrics, which aimed at surveying individual model depths across the whole mantle and retaining geometric features (see also Hosseini et al.[32]).

The tomography models used in this study were chosen to capture a variety of data types and processing techniques that are employed in tomographic inversions. The four P-wave models are DETOX-P01[33], GAP-P4[34,35], HMSL-P06[28], MITP-2011[36], and the four S-wave models are HMSL-S06[28], S40RTS[37], savani[38], SEMUCB-WM1[39]. An earlier vote maps paper[41] applied a similar process to an expanded set of seven P-wave and seven S-wave tomography models. The original models are not filtered to exclude any spherical harmonic degrees, and can be accessed via the SubMachine website[32] http://submachine.earth.ox.ac.uk/.

The individual tomography models are linearly interpolated along a 0.5° grid with a depth increment of 50 km. Processing was done with Generic Mapping Tools (GMT, version 5.3.1[70]). The vote map methodology uses sigma values (standard deviation) as the contour metric for each tomography model at each depth (i.e. ref. [43]). Each model contributes one vote at each grid cell according to whether the model value lies inside (vote = 1) or outside (vote = 0) the specified contour. Supplementary Figure 4 shows an example of the vote map contouring procedure for each tomography model (also applicable to the non-vote maps). Votes for each of the four models are tabulated to generate the final vote map grid as shown in Fig. 5. The resulting abundance profiles (in % of area) of agreement (i.e. how many models agree at a given depth, where four votes is the maximum agreement) for alternative sigma thresholds for the individual seismic tomography models are shown in Supplementary Figs. 2-3. Two additional tomography models TX2019[71] and SP12RTS[72], which show higher variability than the models chosen in our analysis, are included for comparison (Supplementary Fig. 11). They were constructed with different data and with different purposes, to image the core-mantle boundary at long wavelengths for SP12RTS and to test the influence on subducting slabs in the reference model for the TX2019 inversion, which make them less suitable for our focus on the mid-mantle. Scientific, perceptually uniform colour gradients[73,74] were used in all figures.

## Data availability
The data generated in this study have been deposited via Zenodo: https://doi.org/10.5281/zenodo.5519847

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

## Acknowledgements

G.E.S. and R.G.T. acknowledge support from the Research Council of Norway through its Centers of Excellence funding scheme, Project Number 223272. C.H. and J.W.H. were primarily supported by the WPI-funded Earth-Life Science Institute at Tokyo Institute of Technology as well as additional support through JSPS KAKENHI grant numbers 15H05832, 16H06285, 19K04035 and 20K04126. R.M.W. and J.J.V.-C. were funded through NSF grants EAR-1918126 and EAR-2000850. G.E.S. expresses gratitude to Mark A. and Helen M. Shephard.

## Author contributions

G.E.S., C.H., J.W.H., R.G.T. and R.M.W. instigated the study. G.E.S. and J.W.H. undertook the vote map preparation, C.H. and R.M.W. undertook the seismic velocity calculations, R.M.W. and J.J.V.-C. computed elastic moduli and seismic velocities. All authors contributed to the scientific discussion and preparation of the manuscript.

## Competing interests

The authors declare no competing interests.
