## [Peer Review File · Nature Communications]

Seismological Expression of the Iron Spin Crossover in Ferropericlase in the Earth's Lower MantleREVIEWER COMMENTS

Reviewer #1 (Remarks to the Author):

This is very interesting work on the interpretation of seismological tomographic models through mineral properties, specifically the spin crossover in ferropericlase in P-T conditions of the lower mantle. The manuscript is very well written, and very convincing.

There is one weak point, and that is critical: all this is based on the softening of ferropericlase through the spin crossover region. There are several models of such softening published in the mineral physics community. These are reported in the figure below, taken from Marquardt 2018. To my knowledge no consensus as to which model is the correct, and the mineral physics community is still debating this.

The problem here is that the authors use one model for their predictions, and the most extreme one (red points in attached figure). But what happens if the actual F_p in the mantle behave differently? Different models produce very different results.

I don't think the object of this paper is to decide which of these models is the right one to use. It is not the expertise of the authors. For the arguments presented in this paper to hold, they need to be valid for all elastic softening models presented in Marquardt's paper, not just the most extreme one. If the authors can test their seismic interpretation with all these models, they would strengthen their

cannot make the recommendation at this S.

Reviewer #2 (Remarks to the Author):

Review of "Seismological Expression of the Iron Spin Crossover in Ferropericlaase in the Earth's Lower Mantle"

by Shephard, Houser, Hernlund, Valencia-Cardona, Wentzcovitch & Trønnes

Summary

This paper uses an analysis of published mantle tomographic models and mineralogical calculations to assess whether there is evidence for the presence of the spin transition in Fp in Earth's mantle. This is an interesting topic, as the discovery of the Fp spin transition has not been as readily mapped to seismological observations as some other mantle mineralogical phase changes (for example the majorite garnet to Ca-Pv). The paper suggests that the transition is present, at different depths, in both cold and hot parts of the mantle, and can be seen using aggregated velocity maps of fast and slow parts of the mantle, but is not detectable in average 1D models. I raise a few queries here, and otherwise find this to be an interesting paper

Queries raised

- It might be worth referencing more than one seismic model in the introduction, to strengthen your point about the seismic discrepancy - even if this increases the number of references above what n comms normally prefers.
- A comment about the reasons behind the slightly oscillatory nature of some of the Vp profiles in extended figure 1 would be helpful
- Are these models zero mean at every depth, or do any of them have non-zero means at particular depths? If so, how do you deal with the degree zero model contributions
- How do your regions compare with the velocity maps of Cottar & Lekic 2016? I think they had separate Vp and Vs maps, do they have similar changes in the patterns they see, or are the depth levels too spaced out/few?
- The final paragraph suggests that global averaging cannot be responsible for the 'missing' Fp spin transition in 1D models, and that Fp depletion is a possible hiding mechanism. Is this the authors' preferred mechanism, or is it still an open question - ie is this a firm assumption or not.

Minor points

- L10 - clarify that "here" refers to cold regions - I had to re-read to understand
- L10 "a loss in the abundance" loss seems an unexpected word. Consider decrease, or diminution, or something else
- L21 - is it also the case that the thermal variations, as well as chemical ones, lead to the absence of the global signature?
- L46 "mid lower-mantle drop" add a rough indication of depths to make finding this on eg ext figure 1
- L 86 - Extended figure 5 - Consider a different colour scheme - it's not clear to me that red-blue is most intuitive for the differences between vp and vs maps
- L92 - "A drop in slow P-waves relative to slow S-wave anomalies at the base of the mantle" is unclear to me, I think because I have spent less time than the authors thinking about this. I'm not sure if there is another way to phrase it, but it's worth thinking to see if there is.
- In this same journal, two years ago there was an article titled "Valence and spin states of iron are invisible in Earth's lower mantle". The authors may wish to add a comment in the last section of the paper.
- L180 - beyond Shepherd et al 2017, consider also referencing one of Cottar et al 2016 and Lekic et al 2012 (already ref48) papers here.

Reviewer #3 (Remarks to the Author):

General Comments

Compression induced changes in the electron configuration of Fe cations, i.e. Fe²⁺ and Fe³⁺, in mineral phases are one of the most important findings of high-pressure experiments and first-principle computations in a geophysical context. These changes are commonly referred to as spin transitions or spin crossovers and have strong impacts on the elastic properties of the material, in particular by enhancing the compressibility in the pressure and/or temperature interval for which different electron configurations coexist. While the mineral physics community mostly agrees that certain Fe-bearing minerals in Earth's lower mantle will undergo changes in the electron configuration of Fe cations, many details remain to be explored, in particular at combinations of high pressures and high temperatures. In the context of spin transitions, ferroperricite is probably the best-studied mineral. Despite strong experimental and computational evidence in favor of a spin transition of Fe²⁺ in ferroperricite at pressures and temperatures of the lower mantle, the expected impact on the elastic properties of the mantle has not unequivocally been identified in the seismic record and is not apparent in global seismic reference models, such as PREM and ak135. In their manuscript, Shephard et al. search for the signature of the spin transition of Fe²⁺ in ferroperricite in global seismic tomographic models using the technique of vote maps.

The impact of the spin transition of ferroperricite on seismic wave velocities has been analyzed earlier and was found to reduce the sensitivity of P wave velocities to temperature (Wu and Wentzcovitch, 2014). Similarly, vote maps have been used earlier to detect and map the extent of agreement between different seismic tomographic models (e.g. Shephard et al., 2017). Here, the authors apply the technique of vote maps to search for the signature of the spin transition in seismic tomographic models. This approach seems original and promising to me as vote maps emphasize those features that are common to tomographic models as should be any signature of the spin transition if ferroperricite contributes to rocks in the lower mantle with a high enough volume fraction. Indeed, the vote maps constructed by Shephard et al. reveal features that are consistent with the predicted signature of the spin transition of ferroperricite. These features are mostly located in regions of fast seismic velocities that are commonly interpreted as downwelling slabs of upper-mantle rocks. Peridotitic rocks of the upper mantle, i.e. lherzolite and harzburgite, are expected to contain up to about 25 % ferroperricite at pressures and temperatures of the lower mantle. Regions where materials sink from the upper into the lower mantle are therefore expected to be rich in ferroperricite and hence to be characterized by the associated impact of the spin transition on elastic properties. The detection of the spin transition in the seismic properties of regions dominated by fast velocities, i.e. downwelling mantle, by Shephard et al. could be an important step forward in our understanding of deep mantle dynamics as it would link the geodynamic features of downwelling mantle with rock compositions that contain a significant amount of ferroperricite. This finding as well as the application of vote maps to detect seismic signatures of specific rock compositions or phase transitions will be of interest to the entire deep Earth research community. Implications about material exchange between the upper and lower mantle can go beyond the field of mantle geodynamics as they relate to plate tectonic processes and to basaltic volcanism at mid-ocean ridges and hot spots as well as to their associated geochemical characteristics.

Regardless of the successful detection of seismic signatures that are consistent with the spin transition of ferroperricite, I would like to encourage the authors to improve certain aspects of the manuscript, in particular the transparency of the thought process and decision making. In several instances, the line of argument is based on implicit assumptions and/or interpretations that might not be apparent to non-expert readers or might suggest a higher degree of consistency with raised hypotheses than conclusions based on the bare observations. For example, fast seismic velocities are interpreted to be associated with colder temperatures and slow seismic velocities with hotter temperatures without explicitly mentioning this interpretation and without discussing potential trade-offs with rock composition. On a related note, I have the impression that observations seem not always to be clearly separated from interpretations. While experts in the field and readers of more specialist journals might be more familiar with the limitations and caveats of common interpretations and assumptions about seismic tomographic models, I think that a clear separation of observations and interpretations and a transparent line of argument is particularly important in view of the broad readership of Nature Communications.

I would also welcome a more critical assessment of the information content of vote maps. Vote maps reveal common features of tomographic models by filtering for agreement with respect to a defined criterium, i.e. a velocity anomaly "greater than" or "smaller than" a threshold value. Disagreement between models, however, does not imply absence of a feature that has only been observed in a subset of tomographic models. Figures 1b and 1c, for example, show the area coverage of 4/4 votes, i.e. full agreement between the selected models. Lower area fractions of 4/4 votes therefore indicate a higher degree of disagreement between the models at the respective depths, but do not imply that the models agree on the absence of the chosen criterion, i.e. fast P waves in 1b and slow P waves in 1c. In principle, 3 out of 4 models (3/4) could still agree and show fast P wave velocities to follow fast S wave velocities throughout the mantle. To which extent do the outcomes of vote maps depend on the selection of tomographic models? Extended Data Figs. 1 and 2 mainly show the models selected to construct the final vote maps. Why is the loss of fast P wave velocity anomalies in the mid mantle less apparent in tomographic models that combine P and S wave velocities (SP12RTS and TX2019)? Previous mineral-physical models for lower-mantle rocks suggest that P wave velocities are less sensitive to temperature than S wave velocities, even without taking into account spin transitions (e.g. Fig. 4 in Cobden et al., 2009). Temperature-induced P wave velocity anomalies are therefore weaker and more difficult to resolve with seismic tomography than related S wave anomalies. To which extent would the different "background sensitivity" of P and S wave velocities to temperature reduce the observed magnitude (number of votes) of thermally-induced P wave velocity anomalies as compared to S wave velocity anomalies in Figures 3 and 4? Many of the detected P wave anomalies could appear weaker in the vote maps because of the lower temperature-sensitivity of P waves in general, i.e. without need to invoke a spin transition.

In the abstract and introduction, Shephard et al. emphasize the contribution of oceanic lithosphere to downwelling mantle material. Oceanic lithosphere, in particular when "melt-depleted" (line 8), is a complex stack of different lithologies, in the simplest case of basaltic crust and harzburgitic residue that gradually transitions into more fertile lithologies such as lherzolite approaching the composition of the hypothetical rock type pyrolite. The mineral-physical models presented by the authors, however, are exclusively focused on pyrolite. As a consequence, the chosen Fe-Mg exchange coefficients between bridgmanite and ferropericlase (Badro, 2014) are representative of pyrolite with values around 0.5. When exposed to pressures and temperatures of the lower mantle, harzburgite will not only yield higher volume fractions of ferropericlase but also ferropericlase with higher Fe contents due to smaller Fe-Mg exchange coefficients of about 0.2 that decrease further with increasing depth (Piet et al., 2016). Why did the authors decide to model the properties of pyrolite while the spin transition of ferropericlase can be expected to have an even stronger impact on harzburgitic rocks? For pyrolite, thermodynamic models suggest a substantial reduction of the Fe-Mg exchange coefficient with increasing pressure (Nakajima et al., 2012; Xu et al., 2017) that differs significantly from the trend proposed by Piet et al. (2016) and could further enhance the impact of the spin transition by concentrating Fe in ferropericlase. Again, reasons for specific selections and assumptions are not explained in the text.

The following detailed comments mostly point out text sections that support my comments and suggestions as summarized above.

Detailed Comments

Lines 1-2: How well do we know the abundances of ferropericlase and bridgmanite in the lower mantle? As I understand, this paper aims at answering this question. The formula of bridgmanite is highly simplified. What about aluminum and ferric iron?

Line 6: Given the very limited experimental data at combined high pressures and high temperatures on the elastic properties of minerals that undergo spin transitions, I agree with the authors and would expect our models and predictions about spin transitions and their impact on elastic properties to be inaccurate to some degree. This idea is however not further discussed in the manuscript.

Lines 8-10: In contrast to the strong expected impact of the spin transition of ferropericlase on the elastic

properties of "melt-depleted mantle lithosphere of subducted oceanic slabs", the authors present mineral-physical models for pyrolitic compositions only. Oceanic lithosphere, however, is dominated by harzburgite, i.e. pyrolite that has undergone partial melting to form basaltic crust.

Lines 24-25: The statement about the "elusive" composition of the lower mantle is somewhat inconsistent with "90 %" ferropiclasite and bridgmanite (line 1) and the general focus on pyrolitic bulk compositions throughout the manuscript. There seem to be some constraints on the composition of the lower mantle.

Lines 33-34: Why have calculations only been performed for a pyrolitic composition?

Lines 39-40: Regions of fast seismic velocities as observed by Ishii and Tromp (1999) and by Grand (2002) can be interpreted as "cold and sinking oceanic lithosphere" by certainly not identified with such. Again, this is a question to be raised/answered/discussed by the paper. If we already knew that fast anomalies were slabs of oceanic lithosphere composed of peridotite and basalt why search for the spin transition to detect ferropiclasite in these regions?

Lines 44-47: The combined P and S wave models SP12RTS and TX2019 (Extended Data Fig. 3) do not seem to show significant drops in the abundance of fast P wave anomalies in the mid mantle.

Lines 53-56: How well is the "large degree of heterogeneity in tomographic data and modeling approaches" represented in the selection of only 4 S wave and 4 P wave models? Please specify "method" in line 55, i.e. vote maps. Vote maps cannot "cull" inconsistencies between different tomographic models, they rather add up consistencies.

Line 57: I would expect there to be some overlap in the seismic data that contributes to different tomographic models. What if artifacts arise from uneven data coverage or other characteristics of data common to multiple tomographic models? These models could potentially agree on "artifacts" inherited from common data sources.

Lines 58-59: For the sake of transparency and credibility, I strongly suggest to mention and cite selected models and to explain selection criteria in the main text (see lines 167-170).

Lines 59-65: Please specify which "procedure" was used and which criteria applied to construct vote maps. It will be difficult for readers to understand and interpret Figures 1, 3, and 4 without reading the Methods section if no further information is given in the main text on how these figures were created. Vote maps cannot "strip away" features from individual models, they rather compare and filter different models for common features. How were regions of fast and slow anomalies defined in the first place, based on S wave velocities and at which depth? What is meant with "we separate the P- and S-wave signals for fast regions"? How were the signals separated?

Lines 66-68: Figures 1b and 1c cannot unambiguously reveal the "loss of fast P-wave anomalies" as they show the area fractions for 4/4 votes, i.e. for full agreement between all models on the presence (detection) of fast P wave anomalies. A reduction in the area fraction of 4/4 votes merely indicates that the models disagree, but not that they agree on the absence of fast P wave anomalies. Could an additional curve for the area fraction of 0/4 votes, i.e. full agreement on the absence or non-detection of fast anomalies, help to argue for the proposed "loss of fast P-wave anomalies"?

Lines 70-75 Here, I have the impression that the authors implicitly assume fast and slow anomalies to correspond to cold and hot temperatures, respectively, in the mantle. Since factors other than temperature can affect seismic velocities, I suggest to clearly state this assumption before interpreting the observations made on Figure 1c (see also previous comment). Maybe it would be advisable to first describe the observations on regions with overall slow seismic velocities and then interpret or relate these observations to temperature variations and their potential effect on the spin transition.

Line 79: Can a selection of 4 P wave and 4 S wave velocity models represent "every type of tomographic

modelling"? In view of the sparse information on selection criteria provided in the text, I would expect readers to be skeptical about the validity of conclusions drawn from such a fairly specific selection.

Line 81: Please define "coherent structure".

Line 82: In Figure 4, some P wave velocity anomalies appear to become weaker in the mid mantle, but they are certainly not "muted". Other P wave anomalies seem to be clearly resolved by 2-3 tomographic models even in the mid mantle. See also comments on Figure 4 (below).

Lines 83-85: Anomalies of fast seismic wave speeds can be interpreted as or "attributed to" certain lithologies or geodynamic features, but they rarely "show" anything specific. Please separate observations on seismic anomalies from their geodynamic interpretations.

Lines 92-96: LL(S)VPs are characterized by reductions in both S and P wave velocities and an anti-correlation between S wave velocities and bulk sound velocities (see Garnero et al., 2016).

Lines 97-102: I appreciate the discussion of Fe-Mg partitioning or, more generally, Fe-Mg exchange (Badro, 2014) and the effect on the spin transition. However, I suggest to separate the process of partitioning/exchange from the coefficient K that is used to describe this process. It might further be more accessible to readers to explain the compositional broadening of the spin transition in terms of higher Fe contents of ferropericase in response to the spin transition instead of linking the broadening to a decrease of the exchange coefficient K, which is a somewhat indirect expression for and a result of higher Fe contents of ferropericase. In other words, the spin transition broadens and K decreases because of increasing Fe contents in ferropericase. Is there a specific study from which the value $K = 0.5$ has been adopted? First-principles computations and thermodynamic models have suggested that the exchange coefficient decreases systematically across the spin transition, in some models to well below 0.1 (Muir and Brodholt, 2016; Nakajima et al., 2012; Xu et al., 2017). Would such scenarios be expected to have a stronger impact on wave velocities than the Fe-Mg exchange trend suggested by Piet et al. (2016)?

Lines 107-109: Please consider to formulate the observations more precisely: ...drop in the abundance of P wave velocity anomalies compared to the abundance of S wave velocity anomalies...
The second half of the sentence could create the impression that "the ubiquitous presence of Fp in the entire lower mantle" reflects our current view about the composition of the lower mantle, in particular for non-expert readers. I don't think that we know the composition of the lower mantle so well, but Shephard et al. contribute to our picture with an important finding: ferropericase in downwelling mantle regions.

Line 110: Geodynamic models might simplify the complexity of the mantle, but they are certainly not "simple".

Lines 113-116: I have problems in following the line of thought here. What is meant with "combined averages"? I would expect the "inflection in P-wavespeed profiles" to be a result of the spin transition; so this would be an expected rather than a surprising feature when including the effect of the spin transition in forward models of elastic wave speeds.

Lines 116-119: SiO₂-rich or bridgmanite-dominated lower-mantle domains (BEAMS, Ballmer et al., 2017) are indeed a very intriguing possibility to dilute or mask (rather than "suppress") the impact of the spin transition in peridotitic rocks in global average velocity profiles. In the abstract, the authors also mention that currently used models for the spin transition and the associated effects on mineral elasticity might be inaccurate to some extent. Would it be possible to explain this idea a bit more in detail in on one or two sentences?

Line 133: Was ferric iron (Fe³⁺) considered in the modeling of elastic wave speeds? If yes, please add information on the assumed Fe³⁺/ΣFe ratio.

Line 136: Do "self consistent geotherms" imply/mean adiabatic compression paths? In principle, geotherms

describe the actual (but unknown) temperature profile in the mantle.

Lines 167-170: Please consider including this information into the main text. See also comments on lines 58-59 and 59-65.

Figure 1: See comments on lines 66-68. Could additional curves for the area fractions of 0/4 votes, i.e. for full agreement on the absence (strictly non-detection) of fast anomalies, help to argue for the proposed "loss of fast P-wave anomalies" (line 67)? In Figure 1a, there seems to be no pressure axis as mentioned in the caption. The conclusion from Figure 1d in the caption that "a significant departure [from PREM] suggesting that the signature of the spin transition is not a globally-averaged feature" might be difficult to understand from the figure and caption alone. This conclusion implicitly assumes that (or is only valid as long as) 1) S waves are only affected by temperature and 2) the lower mantle has a homogeneous pyrolytic composition (otherwise matching S wave velocities of pyrolyte to PREM to constrain temperature would be of little use). The signature of the spin transition could still be averaged/smeared out in global average models without any of these assumptions being valid.

Figure 3: Please consider changing "fast maps reveal subducted slabs" to "fast velocity anomalies can be interpreted as" and "slow maps reveal" to "slow velocity anomalies as". Also, it would be more precise to state that "P wave velocity anomalies appear less coherent" rather than "P wave model vote maps...". The vote maps are "coherent" in the sense that they cover the entire lower mantle.

Figure 4: The words "coherent" and "coherency" are used in different contexts in the caption; both to describe agreement between tomographic models and to describe the vertical extent of velocity anomalies. This might create confusion for readers. I also suggest to be more precise about fast and slow velocity anomalies instead of referring to "fast vote maps" and "slow vote maps". Why are the vote maps masked for the upper mantle? Including the upper mantle could substantiate the interpretations of fast and slow velocity anomalies in terms of "slabs" and "plumes". In general, P wave velocities are believed to be less sensitive to temperature than S wave velocities, even without a spin transition (e.g. Fig. 4 in Cobden et al., 2009). To which extent could different temperature sensitivities reduce the apparent magnitude (number of votes) of detected P wave velocity anomalies in comparison to S wave velocity anomalies? In Figures 4b and 4c, the overall distribution of velocity anomalies appears to be similar for P and S waves, only that P wave velocity anomalies are weaker, but throughout most of the lower mantle.

References Cited in this Review

- Badro, J., 2014. Spin Transitions in Mantle Minerals. *Annu. Rev. Earth Planet. Sci.* 42, 231–248. <https://doi.org/10.1146/annurev-earth-042711-105304>
- Ballmer, M.D., Houser, C., Hernlund, J.W., Wentzcovitch, R.M., Hirose, K., 2017. Persistence of strong silica-enriched domains in the Earth's lower mantle. *Nat. Geosci.* 10, 236–240. <https://doi.org/10.1038/ngeo2898>
- Cobden, L., Goes, S., Ravenna, M., Styles, E., Cammarano, F., Gallagher, K., Connolly, J.A.D., 2009. Thermochemical interpretation of 1-D seismic data for the lower mantle: The significance of nonadiabatic thermal gradients and compositional heterogeneity. *J. Geophys. Res. Solid Earth* 114, 1–17. <https://doi.org/10.1029/2008JB006262>
- Garnero, E.J., McNamara, A.K., Shim, S.H., 2016. Continent-sized anomalous zones with low seismic velocity at the base of Earth's mantle. *Nat. Geosci.* 9, 481–489. <https://doi.org/10.1038/ngeo2733>
- Grand, S.P., 2002. Mantle shear-wave tomography and the fate of subducted slabs. *Philos. Trans. R. Soc. A Math. Phys. Eng. Sci.* 360, 2475–2491. <https://doi.org/10.1098/rsta.2002.1077>
- Ishii, M., Tromp, J., 1999. Normal-mode and free-air gravity constraints on lateral variations in velocity and density of earth's mantle. *Science* (80-.). 285, 1231–1236. <https://doi.org/10.1126/science.285.5431.1231>
- Muir, J.M.R., Brodholt, J.P., 2016. Ferrous iron partitioning in the lower mantle. *Phys. Earth Planet. Inter.* 257, 12–17. <https://doi.org/10.1016/j.pepi.2016.05.008>
- Nakajima, Y., Frost, D.J., Rubie, D.C., 2012. Ferrous iron partitioning between magnesium silicate perovskite and ferropericlase and the composition of perovskite in the Earth's lower mantle. *J. Geophys. Res. Solid Earth* 117, 1–12. <https://doi.org/10.1029/2012JB009151>
- Piet, H., Badro, J., Nabiei, F., Dennenwaldt, T., Shim, S.-H., Cantoni, M., Hébert, C., Gillet, P., 2016. Spin

and valence dependence of iron partitioning in Earth's deep mantle. *Proc. Natl. Acad. Sci.* 113, 11127–11130. <https://doi.org/10.1073/pnas.1605290113>

Shephard, G.E., Matthews, K.J., Hosseini, K., Domeier, M., 2017. On the consistency of seismically imaged lower mantle slabs. *Sci. Rep.* 7, 10976. <https://doi.org/10.1038/s41598-017-11039-w>

Xu, S., Lin, J.F., Morgan, D., 2017. Iron partitioning between ferropericlase and bridgmanite in the Earth's lower mantle. *J. Geophys. Res. Solid Earth* 122, 1074–1087. <https://doi.org/10.1002/2016JB013543>

Response to Reviewers Round 2: Seismological Expression of the Iron Spin Crossover in Ferropericlasite in the Earth's Lower Mantle

Overview:

We appreciate the thoughtful and constructive comments by the three anonymous reviewers, including two new reviewers. Their comments have driven further improvements to the manuscript. Below we provide a detailed response (plain text) to each comment (*italics*), and describe corresponding changes to text and figures that now appear in the revised manuscript. In our response we use the following abbreviations: Bm = bridgmanite, Fp = ferropericlasite, Vs or S-wave = shear velocity, Vp or P-wave = compressional velocity, Vc = bulk sound velocity.

Highlighted changes:

Clarified justification for compositions used in our study.

Addressed issues raised regarding tomography model resolution and the models used in our study.

Incorporated the various minor issues identified by the reviewers.

Reviewer 3 Comment and Reply:

When reading through the revised version of the manuscript by Shephard et al., I found that the overall organization as well as presentation and discussion of methods and results has been improved substantially. In particular, I appreciate that the authors now include a brief discussion of different rock types associated with the deep recycling of oceanic lithosphere in Earth's lower mantle, and that the analysis of seismic tomographic models is presented in a more transparent and accessible way.

3.1 Reply: Thank you for recognizing our effort, and offering additional helpful comments to further improve the manuscript. We appreciate your continued reviewer feedback.

The information content of the constructed vote maps is explained more explicitly and common assumptions behind the interpretation of fast and slow seismic wave velocity anomalies

are briefly outlined in the introduction, which also contributed to a clearer separation of observations from the authors interpretations.

3.2 Reply: We are glad that our effort achieved its goal.

While it might be worthwhile addressing some persisting minor inconsistencies and ambiguities (see minor comments below), I think that the paper now conveys the authors main message in a comprehensible way and provides sufficient explanation and background information to verify and/or assess the results.

3.3 Reply: It is good to hear that our message is now clearer, we have also addressed these minor comments.

I also would like to suggest going through another round of careful proofreading in order to eliminate any remaining typos.

3.4 Reply: Done.

As pointed out in my previous report, detecting the signature of the spin transition in ferropericlase in the seismic record would be an important step forward in understanding the composition and thermal state of the lower mantle. The findings presented here by Shephard et al. can therefore be considered to be of interest for the entire deep-earth community that connects several disciplines including mineralogy, geophysics, and geochemistry.

3.5 Reply: Thank you for the recognition.

Minor Comments 2.1. In line 25, please consider inserting ferropericlase: increases the compressibility of ferropericlase and decreases its bulk modulus.

3.6 Reply: Done

As mentioned above, I welcome the discussion of different rock types, for example in lines 38 to 42. However, the terms pyrolite, (depleted) peridotite, and harzburgite do not seem to be used in a consistent way. In line 38-39 (and lines 225-226), for example, the authors state that [the] mantle is thought to be a lithological mix of depleted peridotite and [. . .] recycled oceanic crust. Clearly, major parts of the mantle need to be fertile peridotites (pyrolite) to produce basaltic crust at mid-ocean ridges. Why would the authors use a pyrolite model when most of the mantle was actually composed of depleted peridotite? Note that both pyrolite and harzburgite are different types of peridotites. The hypothetical rock pyrolite would correspond to a more fertile and less (melt-)depleted peridotite, e.g. a lherzolite, while harzburgites are peridotites depleted in those components that have been extracted to form basaltic crust.

3.7 Reply: Thank you for highlighting this. We have changed the introduction (lines 43-54) to better explain why we focus on pyrolite in Figures 1 and 2. Pyrolite and peridotite have similar amounts of Fp (17 wt. % versus 18 wt. %) and so the effect of the iron spin

crossover is similar in both compositions. Peridotite represents actual rock compositions exhumed from the upper mantle while pyrolite is a model rock composition. However, pyrolite is the composition most commonly used in comparisons of 1-D seismic models with mineral physics predictions which is why we use it to demonstrate the effects of the iron spin crossover in Fp when its effects are included in the *ab initio* calculations. We have now focused on referring to pyrolite in the main text, but still include peridotite in Table 1 for comparison.

In line 44, it is not clear to me how the Fe/Mg ratio of Fp can be representative for 1-D seismic models. Do you mean consistent with 1-D seismic models? Maybe consider reorganizing this sentence to something like: A pyrolytic model composition with a Fe/Mg ratio of Fp close to 0.19 (16 mole% FeO) has been proposed to be consistent with 1-D seismic models.

3.8 Reply: As the reviewer points out, the proportions mentioned in the original text were distracting. We now refer to pyrolite compositions in general since they are most commonly compared to 1-D seismic models.

In line 51, please consider inserting expected or predicted: within the expected mixed spin region

3.9 Reply: Done

What is meant with different characteristics in line 61? Would it be possible to give examples of such characteristics found in previous comparisons of tomography models in the text?

3.10 Reply: We have modified the sentence (lines 63-67) to "Depending on wavelength and depth, previous comparisons of P- and S-wave tomography models suggest different characteristics in the lower mantle, including in the ratio and correlation of S- and P-wave velocity variations and apparent disruption of imaged fast/slow features..." We have also added the references of Becker and Boschi (2002) and Koelemeijer et al (2018). We also mention the apparent disruption of slabs and plumes later in the discussion section.

In line 75, maybe deviate by more than or similar would be more appropriate than exceed. Slower-than-average wave speeds do not really exceed the median/mean of the velocity distribution

3.11 Reply: Done. We have changed the text to ...deviate by more than....

What is meant with mean of the positive values in lines 117 and 284? Do positive values refer to positive velocity anomalies? Please modify the text accordingly.

3.12 Reply: We have deleted references to contouring around the mean, which is not an important detail here (explanation of the procedures used previously can be found in the cited reference of Shephard et al. (2017)).

In line 139, please replace subducted lithosphere with fast anomalies or similar to keep observations separated from interpretations. The preceding and subsequent sentences consistently describe the vote maps without interpreting them.

3.13 Reply: Done

According to Figure 2c, the temperature-sensitivity of P-wave velocities is not affected by the spin transition at depths in excess of 2500 km. The spin transition can therefore not explain the reduced agreement between tomographic P-wave models in the lowermost mantle (lines 141, 145). See also comment 2.23 on the caption of Figure 6. It might be advisable to specify the depth ranges of what is referred to as mid and lowermost or base of the mantle in the text and to strictly keep the discussion of these two depth ranges separated to prevent miscommunications. In fact, the subsequent paragraph (lines 148-156) seems to focus on the lowermost mantle and offers an alternative explanation (post-perovskite) for observations at depths in excess of 2500 km.

3.14 Reply: Yes, we agree which is why we interpret the observation that the fast and slow Vp and Vs have a similar trend at the base of the mantle as an indication that the crossover to low spin may be mostly complete at these pressures. Also, Fig. 1a shows that the effect of the spin crossover is milder at the highest temperatures near the core-mantle boundary. The results in our study represent the current quantum modeling of the spin crossover, but there are uncertainties in the modeling (discussion of which is beyond the scope of this work) and we plan to investigate the implications at these highest temperatures and pressures in the future. In addition, the post-perovskite transition introduces anti-correlation between S- and P-velocities and separating its signature from that of the spin transition becomes more complicated at the core-mantle boundary.

We now specify the depth ranges we assume when we refer to mid-mantle, base of the mantle, etc.

In line 156, please consider replacing greatly with stronger or similar.

3.15 Reply: We have replaced greatly by more strongly.

In line 160, please consider replacing mole% with molar fraction.

3.16 Reply: We have decided to keep most of the composition references in the text as wt. % consistent with most studies that compare mineral physics predictions to seismological observations for different compositions.

In line 170, please be precise when describing the observations: A reduction in the area fraction of/agreement on/number and extent of detected fast velocity regions

3.17 Reply: The text (line 180) has been changed to "A reduction in the area of mid-mantle fast velocity regions in P-wave models...".

The temperature profile derived from the S-wave velocity match to PREM (line 174) is probably not exactly adiabatic (see also lines 238-240). The conclusion about an adiabatic temperature profile in line 176 can therefore not be based on the comparison of P-wave velocities with PREM as shown in Figure 1d.

3.18 Reply: Thank you for identifying this inconsistency. We meant to indicate that the spin crossover signal cannot “hide” in a Fp-bearing composition such as pyrolite without significant lateral temperature variations. However, a pyrolite composition implies a well-mixed mantle that would be close to adiabatic. We were trying to simplify and reverted to using the term adiabatic without full justification. We have fixed our wording. Also, the reviewer is correct that the obtained temperature profile is not strictly adiabatic but the spin-crossover signature would also be present in a mantle with adiabatic temperatures profiles as shown in, e.g., Refs. 12 and 16.

In line 180, please replace displaces Fe³⁺ from with replaces Fe³⁺ on since atoms are being displaced from their positions by thermal motions but replace each other on crystallographic sites.

3.19 Reply: Done

Since there seem to be very little (experimental) studies on spin transitions at high temperatures in general, I suggest replacing the highest in line 182 with relevant.

3.20 Reply: Done

In lines 186-188, it might be helpful to remind readers why the authors expect certain regions of the mantle to have specific compositions, i.e. recycling and return flow of lithospheric slab materials. What is meant with peridotitic-to-pyrolitic? As noted in comment 2.2, pyrolitic implies peridotitic.

3.21 Reply: We apologize for the confusion. Since we do not know the composition of the lower mantle, we were trying to indicate that peridotitic and pyrolytic compositions both contain enough Fp to expect a seismic signal in these composition if they represent sub-lithospheric mantle or average mantle composition. We have updated the text where either pyrolite or peridotite is mentioned to better explain that they represent compositions where Fp is expected to be present.

It is not quite clear to me how lateral temperature variations can introduce a depth dependence (line 190). Given the reduced temperature-sensitivity of P-wave velocities within the mixed-spin depth interval, I would expect lateral temperature variations to have a smaller effect on global average velocities than without a spin transition.

3.22 Reply: The depth at which the reduction of the bulk modulus due to spin crossover manifests depends on the temperature (see Fig. 1a). Depending on the local geotherm the

P-velocity anomaly will manifest, i.e., have positive/negative depth derivatives, at different depths (as in Fig. 1d). Therefore, this spin crossover signature will be unclear (scrambled) in the global average.

In line 207, please consider replacing Uniquely identifying Fp with Our detection of Fp or Our association of Fp with or similar. How uniquely can a specific mineral be identified by seismic observations on the lower mantle?

3.23 Reply: The text was changed to "...Our detection of Fp...".

In line 223, the authors mention that they included Fe²⁺, Fe³⁺, and Al-bearing bridgmanite in their calculations. What was the Fe³⁺/Fe ratio of bridgmanite? Would it be possible to give the full composition/formula of bridgmanite in the Methods section? At least for the scenario with constant $KD = 0.5$, the mineral compositions should remain constant throughout the considered depth/pressure range. This information would be needed to reproduce the mineral-physical models, for example, to compare the results using mineral elasticity from ab initio calculations with results based on elastic properties derived from high-pressure experiments.

3.24 Reply: Thank you for bringing this to our attention. While addressing this comment, we realized the statement was a result of miscommunication among the authors. We now clarify that Fe and Al were distributed in Bm such that there was no Fe³⁺ in our Bm structure. However, its inclusion would have minimal effect on our velocity calculations (references Shukla, G. et al., GRL, 2016 and Liu, J. et al., Nature communications, 2017) and thus not effect our conclusions regarding the relative behavior of Vs and Vp during the iron spin crossover.

In addition to pyrolite, harzburgite, and the Fp-free composition, Table 1 lists a composition for peridotite that is also mentioned in line 237 but does not seem to be discussed in the main text or included in any of the figures. Both harzburgite and pyrolite are different types of peridotites. Why has this additional composition been added? Does it refer to a specific or natural peridotite composition? In the caption of Table 1, please correct Fe-free to Fp-free.

3.25 Reply: Table 1 caption is now corrected to Fp-free. We have also addressed these issues with the compositions mentioned throughout the text, see 3.7 reply.

In the caption of Figure 2, please correct 1873 km to 660 km.

3.26 Reply: Done

In the caption of Figure 6, please note that the spin transition is not expected to shift to the base of the mantle (penultimate line), even at very high temperatures (see Figure 1a). Please change the caption accordingly.

3.27 Reply: Good point. We replaced the text to: "... shifts to greater depths..."

In Figures 5 and 6, please consider changing the labels number of models to number of votes as in Figure 7. As I understand, all vote maps in Figures 5 and 6 have been constructed from 4 models each.

3.28 Reply: Done.

Reviewer 4 Comment and Reply:

This manuscript provides evidence for the seismological expression of the iron spin crossover in the Earth's lower mantle, based on a comparison between mineral physics calculations and several seismic tomography models. Evidence of a spin transition inside the Earth has profound implications for its convective and thermal evolution. I think this study is probably interesting for the seismology and mineral physics communities because it raises a lot of discussion about how to interpret seismic tomography models and what sort of signals to consider if we want to detect a spin transition using seismology. In that respect it's worth publishing.

4.1 Reply: Thank for recognizing our effort.

Also the argumentation is a bit involved, which can be hard for a non-specialist to follow. Or at least, the way the paper is written and structured could be made cleaner in order to be more compelling. e.g. suddenly discussing iron partitioning after describing 3D seismic observations, then switching to interpretation in terms of 1D structure, is not a comfortable read. Maybe a better structure would be one part of the paper dedicated to the seismic influence of spin transition in terms of F_p abundance, F_e partitioning, etc and another part dedicated to the seismic observations, instead of flip-flopping between them?

4.2 Reply: Thank you for expressing this concern. The iron spin transition is a quantum phenomenon and the lack of an initially obvious seismic expression, especially in 1D seismic models, has led to confusion and/or disregard for its usefulness as a tool to investigate lower mantle composition. The current manuscript structure aims to integrate the predictions, observations, and interpretations throughout the paper. We apologize that this structure is not ideal for the reviewer. Details such as partitioning have been requested by other reviewers and our resulting calculations are a novel contribution to the paper which necessitate discussion before addressing the 1D model predictions and observations. We designed the current text to strike a balance between the complexity of the subject and straightforwardness of the relative V_p and V_s signals. We hope the sum of the edits included in this revised version make the flow easier to follow in general.

The arguments for observing a spin transition are based around the degree of coherency between 4 different P and S tomography models. The issue with using vote maps is it assumes all the tomography models used are equally accurate and their errors are random, when in fact seismic tomography is not a democracy. What if the 1 model who disagrees with the others is the most reliable?

4.3 Reply: Yes, each tomography model has its strengths and weaknesses. Thus, we posit here that signals present in individual models are worth consideration as well as those common to more than one model and evaluate both. We also include many Extended Data Figures to demonstrate how the models effect the fast and slow Vp and Vs signals and allow the audience to examine the feature present in all the individual models.

There is no consideration of resolution/reliability of each model, e.g. with a resolution matrix, to know which bits of each model can be trusted.

4.4 Reply: We agree that it is preferable to evaluate models which are best resolved in the areas of the seismic signal of interest. Here, we focus on the mid-mantle (1200-2500 km) where most models have similar resolution from direct P and S as well as the possible inclusion of other phases such as PP and SS. Ideally, the tomographic inversion should not introduce any structure into the model that is not required by some portion of the data. Thus, we consider the structure in each tomography model to be equally valid for the purpose of this study. The reliability and resolution tests for each individual tomography model are discussed by the authors of those models in the original references. The studies provide checkerboard resolution and other tests which demonstrate their resolving power, and the models considered here have their best resolution in the mid-mantle which we evaluate here.

How much of the greater divergence in P wave models, relative to each other and to S waves, might be due to a lack of ray coverage (and hence more uncertainty) in P? In general Vs is much better constrained than Vp in the mid mantle due to there being more S wave arrivals; does that contribute to the interpretation?

4.5 Reply: It is true that the inclusion of ScS and Sdiff improves the coverage of S-models in model layer at the core-mantle boundary (around 2800 km) and surface waves help to directly constrain Vs in the upper mantle. However, lower mantle seismic velocity variations are mostly constrained by direct P and S, as well as multiples (PP, SS, etc.), which have similar volume and coverage in the mid-mantle.

Could 3D variations in chemistry (or some other physical property) generate the same signal as a spin transition? (if yes, be open about it)

4.6 Reply: To our knowledge, there is no evidence that singular variations in chemistry or temperature or other property could lead to de-correlation between S- and P-velocities at the depths observed in these models. In the abstract and implications section, we openly discuss that any decoupling of Vp and Vs in the mid mantle requires both thermal and chemical heterogeneity across the lower mantle. The signals we observe in fast velocity regions are most consistent with the iron spin transition given our current understanding from mineral physics.

Do all the tomography models have the same 1D reference? Otherwise the definition of fast and slow regions is going to be variable between models, which complicates comparison

with vote maps.

4.7 Reply: Most tomography models are designed/intended to constrain relative variations in seismic velocity, and usually do a poor job of constraining absolute seismic velocities. For example, comparisons of tomography models find that differences in mean/mode of seismic velocity at each depth is not always correlated with differences in 1D reference models (e.g., Hernlund and Houser, EPSL, 2008). Furthermore, the standard deviation of seismic velocity variations vary by up to a factor of 3X, and is strongly affected by subjective decisions such as the kind and amount of damping/regularization. For this reason, we normalize amplitude variations in our analyses.

There is no discussion of a back-transition from low-spin to high-spin iron in the lower-most mantle, yet this is predicted by some mineral physics models, and would appear to be present in some of the profiles plotted in Figure 2? (by the way, next time you submit please can you add the Figure numbers onto the Figures themselves?). Is there any evidence for it in the seismic models?

4.8 Reply: While we can not rule out a back-transition from LS to HS in the deepest mantle, such a back transition is not supported by our calculations. We interpret the increase in surface area covered by fast anomalies in Vp and Vs below 2500 km as the influence of the thermal boundary layer where fast anomalies accumulate and LLSVP dominate the slow anomalies. We consider the observation that the surface area of fast anomalies increases near the base of the mantle to indicate that the coverage/resolution/wavelength ect. of P-wave is capable of resolving cold slabs in the mid-mantle as observed in the fast Vs anomalies, but the expected fast signal from the thermal anomaly is inhibited by iron spin transition.

Also some small things:

Abstract: not sure that ferropericlase is the 2nd most abundant mineral on Earth when its vol% is uncertain. Is there definitely more fp than olivine?

4.9 Reply: We have rephrased the sentence in question. Please see our reply to the next comment.

I would also delete lower mantle from line 1 as there is no such thing as upper mantle bridgmanite.

4.10 Reply: Thank you for the suggestion. The text has been replaced by The two most abundant minerals in the Earths lower mantle are bridgmanite and ferropericlase.

Unless youre looking at anisotropy, references to velocity should be wave speed.

4.11 Reply: Seismic “velocity” is the common term used in seismic tomography. We defer to the recommendations of the editor, and are happy to follow their language/terminology preferences.

Line 194: Im not sure what you mean by Fp concentration being complementary to high viscosity. Correlated with? Associated with?

4.12 Reply: Thank you for pointing out that “complementary” is not clear. We changed the text to: Suppression of a global (1-D) signature of the spin crossover might be explained if significant portions of the lower mantle are depleted in Fp (i.e., enriched in bridgmanite), while Fp-bearing rock that circulates between the shallow and deep mantle is concentrated in upwelling and downwelling channels. This situation is similar to the bridgmanite-enriched ancient mantle structures (BEAMS) model (Ballmer et al 2017) in which Fp-poor regions exhibit a higher viscosity and resist convective mixing.”

Figure 1: the brackets are in the wrong place under step 1 - the) should be after “S-wave” and not “model”

4.13 Reply: Fixed (it was Figure 3).

Figure 4: A legend for the different color lines on the figure itself would be really nice.

4.15 Reply: Amended (the P and S waves do not need to have different colours because they are different panels and we find it is too distracting with more colours)

Reviewer 5 Comment and Reply:

Shephard and co-authors find evidence of a potential mid-mantle signature for the iron spin transition in the mid-mantle by comparing global P and S wave models. I first read the paper and was left with several questions stated below. Reading through the rebuttal did not answer these questions. The previous reviews have clearly improved the manuscript by clarifying many aspects of the methods (including the method outline in figure 3) and potential uncertainties. It also appears that many more figures were added to the supplement for the rebuttal, which in some cases strengthen the case in the paper, but others make the conclusions less convincing (e.g. showing lots of variation within individual models). While I think the approach and analysis are interesting, although the questions below will show I am not entirely convinced. As also mentioned below, the authors could improve convincing a broad audience of the importance of the iron spin transition.

5.1 Reply: Thank you for taking the time to review the manuscript, reviews, reply, and new materials. We appreciate that it is not a small job. We note that there were in fact two rounds of prior responses with this submission to NatComms. The addition of new figures makes the process more transparent, and especially because it shows the variability between the models, which is an important reason why vote maps are utilized, in addition to the individual models.

The authors include 6 different P wave models, 3 of which do not show the expected signature of a spin transition in the area of fast velocities. Two of these (which are two recent models) are only shown at the end of the supplementary material. The authors simply brush

these aside noting these two models were focused on other depths and thus not appropriate to look at the mid-mantle, but such an argument can be made for most models. The authors show that the signal persists for any 3 models picked out of the other 4 in the supplements, but this is not terribly convincing given the previous selection of 4 models.

5.2 Reply: We appreciate the reviewer examining the patterns in each of the six P models. The 2 most-recent models were highlighted in the supplementary material because they have differences from the models used in the main text which make them less suitable for comparison in our analysis. The TX2019 model uses a method that involves accounting for the slab signal in the starting model and SP12RTS is designed for longer wavelength features with emphasis at the core-mantle boundary. Thus, we felt it was important to include them for transparency, but their features are not at odds with the three P models which show a decrease in the surface area of fast velocities in the mid-mantle. Agreement among three P models showing the signal of the spin crossover is worth consideration and further pursuit by others who may find means to better clarify these details. We look forward to future investigation by the wider seismological community.

The authors suggest S and P models have similar resolution across the lower mantle - line 79: 'similar P- and S-wave coverage and thus similar resolution within the midlowermantle'. From my understanding, the HMSL models used contain an extensive ScS data set, but have no equivalent for the P wave model leading to different coverage. Even for comparable coverage, resolution can be different if comparable dominant periods are used, as P waves will have broader Fresnel zones. There is literature showing the unreliability of interpreting these models together, e.g. Tesoniero et al. 2016. Which brings me to my next point. Do the authors have a resolution filter for HMSL to show that slab anomalies with and without spin transition can be well resolved in S and P? Otherwise such filters are available for SP12RTS and have been applied before (Koelemeijer et al. 2018). Of course, it would be better to have such a filter for a model that does show the signal, but demonstrate it is not an artefact.

5.3 Reply: In our manuscript we provide references to two studies that support the argument that S and P are similarly resolved in the mid-mantle (Della Mora et al. 2011 and Moulik and Ekstrom 2016) in addition to the resolution analyses provided in the original tomography model publications. Della Mora et al. (2011) created synthetic travel time data from an artificial Vp model that is a scaled version of Vs. Inverting the synthetic data, they found that the initial scaling of the input model was preserved in the output model. Thus, the data coverage was sufficient in Vp and Vs in the mid-mantle to reproduce the input model without artifacts such as a sudden loss of resolution in fast Vp versus fast Vs at ~1,400 km. Moulik and Ekstrom 2016 graphically demonstrate the depth range over which different types of data in their model are the most sensitive. They show that a combination of even a few different types of body wave data provides sufficient vertical coverage of the mid-mantle. Similar P and S coverage in the mid-mantle is evident in maps of the data coverage and the model error for HMSL Vs and Vp tomography models Houser et al 2008a, and comparable analyses are available for the other models used in our study.

In particular ScS-S improves coverage in the model layer immediately above the core-mantle boundary. Thus, inclusion of ScS-S does not improve coverage in the mid-mantle beyond

that provided by direct S.

The authors support the results of the Tesoniero et al. 2016 study which highlights many issues regarding comparing Vp and Vs in tomography models. Examining the wavelengths of the P- and S-waves would be an interesting avenue of further study. However, we observe that the P-wave data resolve slabs at the top of the lower mantle (down to around 1100 km) and at the base of the mantle below 2500 km, so it seems that the P-wave resolution of slabs is not wavelength dependent but depth dependent.

The authors argue why the spin transition signal is not observed in 1D global profiles, either due to parameterisation or due to a heterogeneous mantle, which makes sense.

5.4 Reply: Thank you for acknowledging this discrepancy.

In lines 198 they state regional profiles are difficult to construct. Why do the authors not consider profiles of absolute velocities extracted from the tomographic models? I.e. in regions that are identified as fast at all depths, interpreted to be slabs, there should be a relative reduction in absolute P velocities and not in S due to the spin transition. This would have to be the case if the reduction in fast area was actually due to an overall drop in velocities there due to the spin transition. This would be a more straightforward demonstration and visual than looking at changes in area. Can the authors demonstrate this, or argue why this wouldn't be the case? (The authors state in the rebuttal to reviewer 1 that 'we can use the seismic observation to constrain the temperature and composition dependence of the iron spin transition'. This seems to imply that the signature should be there in the profiles. The authors also cite papers that have done such analysis for the pv-pvv transition.)

5.5 Reply: Examining the surface area of coverage is one of the main novelties of our study, the signal is much clearer in this measure than any other we have tested. Hernlund and Houser, EPSL, 2008 provides a thorough investigation of the reference 1D models in many P- and S- wave tomography models. It is clear from the previous study that there are no coherent signals in the models that could reveal the patterns associated with the iron spin crossover. In preparation for this study, we examined the 1D signal in the current suite of tomography models and again found no coherent signal.

Looking at the profiles of area of fast velocities in S and P; not only does the P anomaly area decrease, but also the S anomaly area increases (referred to by the authors as a de-correlation between P and S). Is the S wave area increase also attributed to the spin transition?

5.6 Reply: S-waves should not be affected by the spin transition (see Figure 1). Instead, we think the increase in S-wave anomalies reflects an increasing relative abundance of faster material through this depth range. For example, this could be the case if slabs were slowing down and piling up more as they sink into increasingly viscous mantle with depth (this was in fact referred to in Shephard et al. (2017) by looking at slab flux derived from the plate reconstructions and globally averaged mantle sinking rates). The observation that the fast regions resolved by P-waves is opposite that of S-waves is therefore quite unexpected, and is something that appears to be a distinct signature of the spin crossover.

We have added new sentences (lines 91-93) “The overall trend of fast in the S-wave models is one that increases with depth (from 15% coverage at 1200 km depth to 40% coverage near the core-mantle boundary).” ... (lines 156-162) “This could reflect a build-up of slab material and/or a change in the globally averaged subduction flux (Shephard et al. 2017). Furthermore, the area of fast S-wave anomalies increases more rapidly toward the base of the mantle than the P-wave anomalies. This relative difference could (in addition to potential slab-volume changes) be caused by the appearance of post-perovskite in cold mantle, which is expected to increase S-wave velocity by 1-2% but has notably smaller effect on P-waves”.

In the first paragraph, the authors mention the importance of the spin transition, but remain vague about why. Can the authors be more specific as to the effects on dynamics of slabs and plumes? I was left feeling unexcited about the potential finding of the spin transition. It also left me wondering how a transition would affect slabs and plumes and if any effects would show up in tomographic models and in the vote maps?

5.7 Reply: Spin transitions are broad crossovers, not a sharp transition. Nevertheless, they have a “positive Clapeyron slope”. As such, they invigorate convection, irrespective of possible changes in viscosity. It has been argued (Ref. 9) that the bulk modulus softening in the mixed spin region should decrease viscosity in this region, further strengthening convection. Such argument has been used in numerous geodynamic simulations, e.g., Bower, D. J., Gurnis, M., Jackson, J. M. Sturhahn, W. Enhanced convection and fast plumes in the lower mantle induced by the spin transition in ferropericlase. *Geophysical Research Letters* 36, doi:10.1029/2009gl037706 (2009).

We find the observation of the iron spin transition which reveals the presence and distribution of Fp in the lower mantle to be the fundamental excitement regarding our observations. The studies we reference more fully describe all the possible implications of the iron spin transition.

Figure 4: The fast and slow areas show massively different behaviour in P and S at the top of the lower mantle 800-1000 km, where S increases for fast and P for slow. Apologies if I missed any discussion on this. The area profiles for the individual models in the supplement show this signal is more robust than that claimed in the mid-mantle across the different models.

5.8 Reply: We agree that this is a very interesting signal, and deserves further investigation with respect also to the transition zone. However, we believe that this is beyond the scope of the present study. We also defer more discussion about behaviours in the lowermost mantle to later studies.

Figure 6 - the cross-sections show a strange kind of ringing in the number of votes, not seen in the maps (looks like crystal zoning?)? Is this a plotting artefact?

5.9 Reply: This is a function of plotting (done with with Generic Plotting Tools (GMT)) which is achieved by creating a vertical grid of data from horizontal (depth) datasets. The various models are analysed with set increments/grid nodes and it is expected that there will

be variability switching from vote/model counts over these increments. Also, the contouring method applies a threshold based on the standard deviation so the apparent ringing is not likely a real dynamic observation and just highlights the overall geometry of the mantle structures. We could apply an additional smoothing filter to remove this but feel this would add an extra level of obscurity to the a fair interpretation.

Suppl. Figure 11, please clarify what the mean models are plotted with respect to (PREM?).

5.10 Reply: They are not necessarily plotted with respect to PREM (depends on the individual model construction; see also Shephard et al., 2017 and Hosseini et al., 2018). Nonetheless, PREM is the reference model for S40RTS and savani. We have clarified this to “Panels a and b reveal the depth-dependent mean and standard deviation for each of the 8 individual models used in this study, respectively, before any gaussian fitting is applied. The models are plotted with respect to their original model construction (see Hosseini et al., 2018 the SubMachine website for further model-specific details).”

REVIEWER COMMENTS

Reviewer #3 (Remarks to the Author):

1. Overview

By going through a second round of revision, Shephard et al. have further improved the organization and presentation of their manuscript that aims at detecting the seismic signature of the spin transition in the mineral ferropericlasite in Earth's lower mantle. I think that manuscript now follows a coherent line of thought and that the results will be of interest to the wider deep-earth research community. The description of the selection and analysis of seismic tomographic models and their combination into vote maps presents sufficient detail to be accessible for verification and to serve as a blueprint for future studies, for example including other or more seismic tomographic models. The detection of disparate variations in the area fractions of P and S wave speed anomalies with depth and the overall consistency of these variations with characteristics of the spin transition as predicted by mineral physics will be seen as an important step forward across the deep-earth research community. Although I have a few last comments and suggestions, I think that these would be easy to address before publication and, from my point of view, would not require another round of review.

As asked for by the editor, I would further like to mention that I think the authors have adequately addressed the comments and concerns raised by Reviewer #4. In particular, I see that most concrete suggestions have been incorporated while more general concerns have either been discussed in the reply letter or did not seem fully justified to me. For example, Reviewer #4 asked about the "definition of fast and slow regions" in view of different seismic reference models that are employed to construct different seismic tomographic models. In lines 80–83 and in the Methods section "Fast and slow mapping", however, the authors explain in detail how they rescaled seismic wave speed anomalies of different seismic tomographic models to establish a common criterion for "fast" and "slow" that is applicable to each seismic tomographic model, regardless of the reference model. For comments about characteristics of individual seismic tomographic models, e.g. resolution, accuracy, etc., I think it is important to keep in mind that Shephard et al. do not intend to argue for or against specific seismic tomographic models or to present a review; here the authors search for features that several seismic tomographic models have in common. Each of the seismic tomographic models has been published as a separate contribution, passed peer-review, and can be assessed based on the respective published information. In addition to the formal reply in the rebuttal letter, I think that the comment of Reviewer #4 about alternative explanations ("3D variations in chemistry or some other physical property") for the detected seismic signature, which the authors attribute to the spin transition, could have been addressed by adding a comment to the text, for example towards end of the implications section around lines 214–215, to acknowledge that other factors could in principle generate a similar seismic signature.

2. Major Comments

2.1. To define fast and slow wave speed anomalies, the authors analyze the "seismic velocity distribution over a reference depth range of 1000–2200 km" (lines 82–83) to derive standard deviations. Wave speed variations that deviate by more than one standard deviation from the centroid of this distribution are classified as "anomalies" (see also Methods section "Fast and slow mapping", lines 260–286). Strictly, this definition will only be valid within the selected depth range. At depths shallower than 1000 km or deeper than 2200 km, the wave speed distributions might be different and be characterized by different standard deviations. In particular, such changes in the S wave speed spectrum have been identified at around 1000 km and for the D" layer (Durand et al., 2017, 2016) and are acknowledged by the authors in lines 272–273. The limitations of applying a criterion defined for anomalies at depths of 1000–2200 km to detect anomalies at shallower/deeper depths might be worth mentioning in the main text, in particular when discussing observations "near the core-mantle boundary" (line 93), on the "lowermost mantle" (lines 148–149, 153), the "base of the mantle" (line 159), or similar. For example, the increase in the area fractions of anomalous regions "between ~2500–2800 km depth" (line 157) could simply result from broader velocity distributions at these depths when compared to those of the mid mantle, i.e. between 1000 and 2200 km. Similarly, the "dominance of S-wave votes" (lines 152–156, see also lines 158–159) in the lowermost mantle might indicate that S wave speeds show a wider spectrum than P wave speeds. I suggest including a statement

that clearly emphasizes the extrapolative character of the definition of anomalies when applied to depths outside/beyond the depth interval used to define the reference wave speed distributions and to exert caution when interpreting the respective area fractions of anomalous regions. Such a statement could be inserted around line 157, for example. The detection of a seismic signature that is consistent with the spin transition in the mid mantle would remain untouched from this concession.

3. Minor Comments

3.1. In line 2, "d-electron configuration" might be more appropriate since individual electrons cannot transition into a high-spin or low-spin state, which are always multi-electron states.

3.2. In lines 10, 16, 40, 186, 191, 196, 209 (and potentially elsewhere), the authors refer to "1-D seismic profiles". It might be more appropriate to refer to "global seismic reference/average profiles" or similar to emphasize the global or average character of these reference profiles as opposed to "regional 1-D profiles" (lines 212) or at least to clearly define at the beginning of the manuscript what is meant with "1-D seismic profiles".

3.3. In line 19, please consider changing to: ... it is expected to alter ...

3.4. As mentioned in my previous evaluation of this manuscript, I strongly suggest to unambiguously define mineral compositions in terms of molar fractions, chemical formulae, or elemental ratios, i.e. Fe/(Mg+Fe), Fe/Mg ratios. Statements like "iron concentrations of ~20%" (line 34), "15% iron concentration" (lines 45, 252, captions of Fig. 2 and Supp. Fig. 8) are prone to misinterpretation. Please revise accordingly.

3.5. In line 34, please consider changing to: ... which is believed to be the case ...; higher iron concentrations, or Fe/Mg ratios, have been invoked to explain features of LLVPs and ULVZs, for example.

3.6. In line 47, please correct to "reveal".

3.7. In line 64, please consider inserting "seismic": ... different seismic characteristics ...

3.8. In line 64, please consider inserting "changes": ..., including changes in ...

3.9. It is not clear to me how the "change in the globally averaged subduction flux" (line 158) can be derived from the analysis of seismic models. Where does the time constraint come from?

3.10. In line 168, it might be worthwhile adding a citation to direct interested readers to references about "core-mantle boundary layer dynamics".

3.11. In line 198, please consider inserting "recycled" or "deeply subducted": ... detectable in the recycled Fp-enriched ...; oceanic lithosphere does not contain Fp unless it is being subducted into the lower mantle.

3.12. As mentioned in my previous report, the phrase "lateral temperature variations introduce a depth dependence" (lines 200–201) might potentially be confusing to some readers: how can a lateral variation give rise to a depth dependence? From the authors' reply, I understand they want to say that lateral temperature variations will tend to dilute/spread/smear/blur the signature of the spin transition in the vertical dimension. Please consider rephrasing this sentence accordingly.

3.13. In line 238, please correct to "thus do not".

3.14. Please note that experiments indicate a measurable difference in sound wave velocities between ferric iron free and ferric iron bearing bridgmanite (Kurnosov et al., 2017). Maybe adding "in ferropericlase" at the end of the sentence in line 239 would help to remind readers about the focus of the paper on the spin transition in ferropericlase as opposed to bridgmanite.

3.15. In line 316 and in the captions of Figures 1 and 5, the term "colour maps" might lead to confusion with the maps displayed in figures; please consider using "colour scales" or "colour gradients" instead.

3.16. In the caption of Figure 4, please consider using the term "bold lines" instead of "dark lines" since there seems to be gradient from lighter to darker colors for different tomographic models.

3.17. In the caption of Figure 5, panel a is associated with S waves and panel b with P waves. In the figure, however, the situation seems to be reversed. Please double-check and correct caption or figure accordingly.

3.18. The information in the first sentence of the caption of Table 1 is not related to the content of the table. Please remove this sentence to keep the caption as concise as possible.

3.19. Would it be possible to plot the temperature profile that has been derived by matching S waves to PREM in Figure 1a and/or in Supplementary Figure 9? Since the inflexion of P wave speeds in Figure 1c appears more intense than in the coldest scenario shown in Figure 2c, it would be interesting to see the corresponding thermal profile.

3.20. It is not quite clear to me what is being plotted in Supplementary Figure 7. What is meant with the notation "4+3 votes" etc.? Would it be possible to rephrase the caption to better explain what is being shown in the figure?

3.21. From lines 80–83 in the main text and lines 270–272 in the Methods section, I understand that all wave speed variations between 1000 and 2200 km depths have been combined to create the mid mantle reference distributions for each seismic tomographic model, i.e. one distribution for S waves and one for P waves per model, which are then analyzed by Gaussian fits as shown in Supplementary Figure 10. The caption of Supplementary Figure 10, however, states that the authors “analyze the Gaussian portion of the models at each depth”. The last sentence of the figure caption is somewhat confusing in general. Please revise the figure caption as needed.

3.22. What is plotted along the horizontal axes in Supplementary Figures 11a, 11b, and 11c? Do the numbers refer to wave speed variations in percent as reported in the original seismic tomographic models? Please add more explanatory axis labels and units as needed. It is also not entirely clear what is plotted in Supplementary Figure 11c. In principle, the Gaussian fitting of reference wave speed distributions should yield a single standard deviation for the entire depth range 1000–2200 km for P and S waves, respectively, and for each model. The “recalculated/filtered standard deviations” shown in Supplementary Figure 11c, however, are depth dependent. Please explain in the figure caption. I also suggest changing “Extended Figure 10” to “Supplementary Figure 10” for consistency.

4. References

Durand, S., Debayle, E., Ricard, Y., Lambotte, S., 2016. Seismic evidence for a change in the large-scale tomographic pattern across the D'' layer. *Geophys. Res. Lett.* 43, 7928–7936.

<https://doi.org/10.1002/2016GL069650>

Durand, S., Debayle, E., Ricard, Y., Zanolli, C., Lambotte, S., 2017. Confirmation of a change in the global shear velocity pattern at around 1000km depth. *Geophys. J. Int.* 211, 1628–1639.

<https://doi.org/10.1093/gji/ggx405>

Kurnosov, A., Marquardt, H., Frost, D.J., Boffa Ballaran, T., Ziberna, L., 2017. Evidence for a Fe³⁺-rich pyrolitic lower mantle from (Al,Fe)-bearing bridgmanite elasticity data. *Nature* 543, 543–546.

<https://doi.org/10.1038/nature21390>

Reviewer #5 (Remarks to the Author):

The authors have improved the paper further, and it should be close to publishable.

I enjoyed reading the responses from the authors, and hearing more about the analyses that went in. However, only in a few locations did my thoughts/confusions lead to improvements of the paper, and others were brushed aside. The authors have the right to do this, although I do think future readers might have similar concerns. However, I am happy for some of this discussion to continue after the paper is published.

There are two points where I was a bit more disappointed to not see changes.

- The authors did not include more background on the implications of the spin transition. This could either go as motivation in the introduction, or it would also be nice if the paper ended with implications. A fraction of the broad set of disciplines this paper caters to, will not be familiar.

- The ringing in Figure 6 isn't fixed. I would think there would be ways around this... i.e. interpolating vote values between grid points (it can still be plotted on a discrete colour scale after).

In both cases I will leave it up to the editor if these should be implemented before publication.

3 August 2021

We thank the two reviewers again for their time and dedication to reviewing this manuscript, and their careful and useful comments.

Reviewer 3 on behalf of Reviewer 4.

Overview comment: “In addition to the formal reply in the rebuttal letter, I think that the comment of Reviewer #4 about alternative explanations (“3D variations in chemistry or some other physical property”) for the detected seismic signature, which the authors attribute to the spin transition, could have been addressed by adding a comment to the text, for example towards end of the implications section around lines 214–215, to acknowledge that other factors could in principle generate a similar seismic signature.”

Reply: Amended. The sentence at 218 now reads “*It is possible that more complicated combinations of thermal, phase, and/or chemical composition variations could provide an alternate explanation for Vs-Vp decorrelation. However, the pressure-temperature dependence of the Fp spin crossover provides a unified explanation for why this occurs at these particular distinct depths in slow and fast regions, and does so without the need to invoke depth-dependent changes in the chemical composition of downwelling and upwelling materials..*”

As a summary to the main text, we have also added to the Methods (line 292) “*The differing extent of mid-mantle fast anomalies in Vp and Vs requires the simultaneous effect of temperature, composition, and phase. Any temperature effect alone would be expected to manifest in Vs and Vp to a similar extent and magnitude. Since fast anomalies are often interpreted as cold subducted oceanic lithosphere, the next property to consider is composition. Subducted slabs are composed of a thin basaltic crust and a Fp-rich mantle lithosphere. The thin basaltic crust may host anomalous seismic characteristics, but these seismic anomalies are not expected to be detected by seismic tomography at scales of 1000’s of km in the mid-mantle. If Fp exists in higher quantities in fast velocity regions, then the bulk rock would have slightly reduced velocity since Fp has slower Vp and Vs than Bm. However, the fast velocity anomaly observed in seismic tomography indicates that temperature dominates the seismic signal even in these more Fp-rich subducting slabs. Since variations in Fp concentration have a similar effect on Vp and Vs, it is difficult to decouple Vp and Vs even with temperature and composition. Fp concentration in slabs opens up the possibility of the iron spin transition which reduces Vp and not Vs in the presence of a thermal anomaly.*”

Major comment 2.1 To define fast and slow wave speed anomalies, the authors analyze the “seismic velocity distribution over a reference depth range of 1000–2200 km” (lines 82–83) to derive standard deviations. Wave speed variations that deviate by more than one standard deviation from the centroid of this distribution are classified as “anomalies” (see also Methods section “Fast and

slow mapping”, lines 260–286). Strictly, this definition will only be valid within the selected depth range. At depths shallower than 1000 km or deeper than 2200 km, the wave speed distributions might be different and be characterized by different standard deviations. In particular, such changes in the S wave speed spectrum have been identified at around 1000 km and for the D” layer (Durand et al., 2017, 2016) and are acknowledged by the authors in lines 272–273. The limitations of applying a criterion defined for anomalies at depths of 1000–2200 km to detect anomalies at shallower/deeper depths might be worth mentioning in the main text, in particular when discussing observations “near the core-mantle boundary” (line 93), on the “lowermost mantle” (lines 148–149, 153), the “base of the mantle” (line 159), or similar. For example, the increase in the area fractions of anomalous regions “between ~2500–2800 km depth” (line 157) could simply result from broader velocity distributions at these depths when compared to those of the mid mantle, i.e. between 1000 and 2200 km. Similarly, the “dominance of S-wave votes” (lines 152–156, see also lines 158–159) in the lowermost mantle might indicate that S wave speeds show a wider spectrum than P wave speeds. I suggest including a statement that clearly emphasizes the extrapolative character of the definition of anomalies when applied to depths outside/beyond the depth interval used to define the reference wave speed distributions and to exert caution when interpreting the respective area fractions of anomalous regions. Such a statement could be inserted around line 157, for example. The detection of a seismic signature that is consistent with the spin transition in the mid mantle would remain untouched from this concession.

Reply: We completely agree with the reviewer. We have added a sentence to the at line 155 that reads as follows: “*Note that our definition of anomalous material is based on characteristic behaviors in the depth range 1,000-2,200 km (see Methods), and any discussion of anomalies outside of this depth range should be considered relative to these standards.*”

3. Minor Comments

3.1. In line 2, “d-electron configuration” might be more appropriate since individual electrons cannot transition into a high-spin or low-spin state, which are always multi-electron states.

Reply: Amended. It now reads “*Mineral physics experiments and theory consistently predict that the d-electrons of Fe^{2+} in Fp , $(Mg,Fe)O$, change from a high-spin to low-spin (HS-LS) state at mid-lower mantle conditions (Fig. 1).*”

3.2. In lines 10, 16, 40, 186, 191, 196, 209 (and potentially elsewhere), the authors refer to “1-D seismic profiles”. It might be more appropriate to refer to “global seismic reference/average profiles” or similar to emphasize the global or average character of these reference profiles as opposed to “regional 1-D profiles” (lines 212) or at least to clearly define at the beginning of the manuscript what is meant with “1-D seismic profiles”.

Reply: Amended. The first instance in the main manuscript (line 16) now reads “*However, the compressional velocity of a homogeneous pyrolitic mantle (Fig. 1d) does not fit global seismic average profiles when the effects of the iron spin*

crossover are included in the predicted seismic velocity computations[ref]” And the second instance (line 41) is clarified with “While a pyrolytic model compositions have been proposed to be consistent with 1-D seismic models [ref] (i.e. *global seismic reference/average profiles*), Fig. 2 demonstrates...”

3.3. In line 19, please consider changing to: ... it is expected to alter ...

Reply: Amended.

3.4. As mentioned in my previous evaluation of this manuscript, I strongly suggest to unambiguously define mineral compositions in terms of molar fractions, chemical formulae, or elemental ratios, i.e. Fe/(Mg+Fe), Fe/Mg ratios. Statements like “iron concentrations of ~20%” (line 34), “15% iron concentration” (lines 45, 252, captions of Fig. 2 and Supp. Fig. 8) are prone to misinterpretation. Please revise accordingly.

Reply: Agreed. We now report compositions in a clearer style. Table 1 and caption have been modified as requested. All compositions throughout the paper are now reported in a similar manner and the ferropericlase compositions in the main text and figure captions are referred to as (e.g. in line 36) “--- for FeO concentrations below 20 mol% in Fp, ---”.

3.5. In line 34, please consider changing to: ... which is believed to be the case ...; higher iron concentrations, or Fe/Mg ratios, have been invoked to explain features of LLVPs and ULVZs, for example.

Reply: Amended. It now reads “Specifically, the transition pressure does not change significantly for FeO concentrations below 20 mol% in Fp, which is representative for the lithological range from fertile peridotite to harzburgite.”

3.6. In line 47, please correct to “reveal”.

Reply: Amended.

3.7. In line 64, please consider inserting “seismic”: ... different seismic characteristics ...

Reply: Amended.

3.8. In line 64, please consider inserting “changes”: ..., including changes in ...

Reply: Amended.

3.9. It is not clear to me how the “change in the globally averaged subduction flux” (line 158) can be derived from the analysis of seismic models. Where does the time constraint come from?

Reply: Amended. It is from the application of a globally averaged sinking rate (e.g. 1cm/yr). It now reads *“This could reflect a build-up of slab material and/or a change in the globally averaged subduction flux, as estimated from mantle sinking rates [ref]”* we also added a reference to van der Meer et al. (2010).

3.10. In line 168, it might be worthwhile adding a citation to direct interested readers to references about “core-mantle boundary layer dynamics”.

Reply: Amended. We now reference Garnero et al. (1998), Cottaar and Lekic (2016) and Heyn et al. (2020).

3.11. In line 198, please consider inserting “recycled” or “deeply subducted”: ... detectable in the recycled Fp-enriched ...; oceanic lithosphere does not contain Fp unless it is being subducted into the lower mantle.

Reply: Amended.

3.12. As mentioned in my previous report, the phrase “lateral temperature variations introduce a depth dependence” (lines 200–201) might potentially be confusing to some readers: how can a lateral variation give rise to a depth dependence? From the authors’ reply, I understand they want to say that lateral temperature variations will tend to dilute/spread/smear/blur the signature of the spin transition in the vertical dimension. Please consider rephrasing this sentence accordingly.

Reply: Amended (line 205) to *“Variations of lateral temperature with pressure introduces a depth-dependence to the P-wave seismic velocity inflection (i.e. causing a potential vertical smearing of the signature), which could reduce the globally averaged signal”*

3.13. In line 238, please correct to “thus do not”.

Reply: Amended.

3.14. Please note that experiments indicate a measurable difference in sound wave velocities between ferric iron free and ferric iron bearing bridgmanite (Kurnosov et al., 2017). Maybe adding “in ferropericlase” at the end of the sentence in line 239 would help to remind readers about the focus of the paper on the spin transition in ferropericlase as opposed to bridgmanite.

Reply: Amended. And added also sentence *“Here we consider the effects of a spin crossover in Fp, proposed spin changes in Fe in Bm are not considered.”*

3.15. In line 316 and in the captions of Figures 1 and 5, the term “colour maps” might lead to confusion with the maps displayed in figures; please consider using “colour scales” or “colour gradients” instead.

Reply: Amended.

3.16. In the caption of Figure 4, please consider using the term “bold lines” instead of “dark lines” since there seems to be gradient from lighter to darker colors for different tomographic models.

Reply: Amended.

3.17. In the caption of Figure 5, panel a is associated with S waves and panel b with P waves. In the figure, however, the situation seems to be reversed. Please double-check and correct caption or figure accordingly.

Reply: Amended

3.18. The information in the first sentence of the caption of Table 1 is not related to the content of the table. Please remove this sentence to keep the caption as concise as possible.

Reply: Amended

3.19. Would it be possible to plot the temperature profile that has been derived by matching S waves to PREM in Figure 1a and/or in Supplementary Figure 9? Since the inflexion of P wave speeds in Figure 1c appears more intense than in the coldest scenario shown in Figure 2c, it would be interesting to see the corresponding thermal profile.

Reply: We have added a plot to Supplementary Figure 9 and amended the caption accordingly:

“Top panel: Development of Figure 1d. PREM is shown in black circles and the black lines are the calculated velocities for pyrolite⁸. Figure 1d demonstrates the spin transition effect on V_p for the case in which predicted V_s matches PREM (grey lines). Since V_s for pyrolite does not fit PREM with an adiabatic temperature gradient^{9, 10}, the temperature profile that shifts V_s to align with PREM (grey line right panel) undulates in the lowermost mantle. Bottom panel: The self-consistent geotherms from our pyrolite calculations⁸ for the elastic moduli and velocity profiles plotted in Figure 2. The calculations start by setting the temperature at the top of the lower mantle to 1373 K (blue, the -500 K case), 1873 K (black, the average case), and 2373 K (red, the +500 K case) and allowing the temperature to increase adiabatically as the calculations proceed to higher pressures across the lower mantle.”

3.20. It is not quite clear to me what is being plotted in Supplementary Figure 7. What is meant with the notation “4+3 votes” etc.? Would it be possible to rephrase the caption to better explain what is being shown in the figure?

Reply: Rephrased to “In Figure 1, only the area corresponding to the maximum vote of 4 was shown; “4+3 votes” indicates that the area corresponding to votes of 3 and 4 are summed and plotted, “4+3+2” indicates votes of 2, 3 and votes are summed and plotted etc.”

3.21. From lines 80–83 in the main text and lines 270–272 in the Methods section, I understand that all wave speeds variations between 1000 and 2200 km depths have been combined to create the mid mantle reference distributions for each seismic tomographic model, i.e. one distribution for S waves and one for P waves per model, which are then analyzed by Gaussian fits as shown in Supplementary Figure 10. The caption of Supplementary Figure 10, however, states that the authors “analyze the Gaussian portion of the models at each depth”. The last sentence of the figure caption is somewhat confusing in general. Please revise the figure caption as needed.

Reply: We thank the reviewer for catching this confusing description. We amended caption of Supplementary Figure 10 to:

“The results of our Gaussian-fitting procedure (see Methods) for all 8 tomography models used in this study. Analysis of velocity-frequency distributions of a variety of tomographic models reveals that they exhibit significant differences that confound inter-model comparisons (Hernlund and Houser, EPSL, 2008). These differences can be categorized as scale/amplitude (e.g., caused by variability in tomographic model data, design, regularization), shift/alignment (e.g., caused by reference to different 1-D global models), and shape of the distributions (variations in distribution morphology that remain even after accounting for linear shift and scale differences). By analyzing distributions we find that all models yield Gaussian-like variations in V_p and V_s in the depth range 1,000-2,000 km, however, there are particularly large discrepancies in amplitude between the different models (Hernlund and Houser, EPSL, 2008). These scale differences must be normalized to a reference standard in order to establish a useful definition for fast and slow anomalies that can be compared across the suite of models. We do this by combining each model from 1,000-2,200 km depth, and performing iterative Gaussian fitting to the central portion (i.e., within $\pm\sigma$) of the resultant distribution as described in Methods. The value of σ obtained in this manner is then used to define what qualifies as fast and slow anomalies in the models.”

3.22. What is plotted along the horizontal axes in Supplementary Figures 11a, 11b, and 11c? Do the numbers refer to wave speed variations in percent as reported in the original seismic tomographic models? Please add more explanatory axis labels and units as needed. It is also not entirely clear what is plotted in Supplementary Figure 11c. In principle, the Gaussian fitting of reference wave speed distributions should yield a single standard deviation for the entire depth range 1000–2200 km for P and S waves, respectively, and for each model. The “recalculated/filtered standard deviations” shown in Supplementary Figure 11c, however, are depth dependent. Please explain in the figure caption. I also suggest changing “Extended Figure 10” to “Supplementary Figure 10” for consistency.

Reply: We agree with the reviewer that this figure and caption is confusing, and this information is anyways not directly relevant to the procedures we employed in this paper. Furthermore, this kind of information has already been analyzed

and discussed in prior publications, which are referenced numerous times in this manuscript. Therefore, we have decided to remove Supplementary Figure 11 altogether.

Reviewer 5.

There are two points where I was a bit more disappointed to not see changes.
- The authors did not include more background on the implications of the spin transition. This could either go as motivation in the introduction, or it would also be nice if the paper ended with implications. A fraction of the broad set of disciplines this paper caters to, will not be familiar.

Reply: In the interests of brevity for the Nature Comms format we have modified the sentence at line 21 to read "*The rheological consequences of such material changes mean that subducted slab, mantle plume, and deep mantle dynamics are thought to be affected by the crossover, including enhanced mixing and slab stalling* \cite{Shahnas17,Bower09}."

The ringing in Figure 6 isn't fixed. I would think there would be ways around this... i.e. interpolating vote values between grid points (it can still be plotted on a discrete colour scale after).

In both cases I will leave it up to the editor if these should be implemented before publication.

Reply: We have found that including the different levels of vote map agreement when plotting depth slices and cross sections makes it possible for the readers to see for themselves how many models agree on geographic extent of the fast/slow velocity anomaly patterns. We believe that interpolating the vote map counts would unnecessarily complicate our otherwise straightforward plotting procedure.